# DoseSurv: Predicting Personalized Survival Outcomes under Continuous-Valued Treatments

**Moritz Gögl**[1]   **Yu Liu**[1]   **Christopher Yau**[1,2]   **Peter Watkinson**[1]   **Tingting Zhu**[1]

[1]University of Oxford
[2]Health Data Research UK
moritz.gogl@keble.ox.ac.uk

## Abstract

Estimating heterogeneous treatment effects (HTEs) of continuous-valued interventions on survival, that is, time-to-event (TTE) outcomes, is crucial in various fields, notably in clinical decision-making and in driving the advancement of next-generation clinical trials. However, while HTE estimation for continuous-valued (i.e., dosage-dependent) interventions and for TTE outcomes have been separately explored, their combined application remains largely overlooked in the machine learning literature. We propose DoseSurv, a varying-coefficient network designed to estimate HTEs for different dosage-dependent and non-dosage treatment options from TTE data. DoseSurv uses radial basis functions to model continuity in dose-response relationships and learns balanced representations to address covariate shifts arising in HTE estimation from observational TTE data. We present experiments across various treatment scenarios on both simulated and real-world data, demonstrating DoseSurv's superior performance over existing baseline models.

## 1 Introduction

Estimating the causal effects of interventions at the individual level is essential for informed decision making in fields such as personalized medicine [48], public health [39], social and economic policy [62], and marketing and retail [44]. Within the machine learning (ML) literature, predominant attention has been directed toward estimating heterogeneous treatment effects (HTEs) on continuous and binary outcomes [1, 2, 62, 70, 63, 60, 9, 16]. By contrast, time-to-event (TTE) outcomes, which capture the time until an event of interest, have received comparatively less attention. More specifically, the existing ML literature on HTE estimation with TTE outcomes has focused on quantifying causal effects of binary (non-dosage) interventions with "treatment" or "no treatment" (control) being the only options [14, 58]. However, in practice, treatment settings are typically more complex, comprising several, possibly continuous-valued treatment options, characterized by their dosage or frequency of administration.

TTE outcomes have immense practical relevance, for example, in clinical studies that analyze survival time or time to discharge. Equally, studying the causal effects of continuous-valued treatments is an important problem in healthcare, with relevant areas including chemotherapy [18], insulin regimens [36], and cardiovascular treatments [21]. Underlying dose-response relationships can be complex and non-monotonic, with multiple stimulatory and inhibitory phases. Although average dose-response curves across a population can be characterized in randomized controlled trials (RCTs) and may appear simple, dose-response relationships at the patient level are often more intricate. Individual patient characteristics can complicate these relationships, leading to patient-specific ideal dosages and necessitating continuous modeling of treatments at the individual level.

While neural network (NN) models for HTE estimation have been developed separately for continuous-valued interventions [60, 9, 51] and for TTE outcomes [12, 14, 58], their joint application remains

largely unexplored in the current ML literature. Therefore, we propose *DoseSurv*, a deep learning survival model that estimates the conditional hazard function–the probability of experiencing the event at a given time–under different treatments and dosages. Unlike many (semi-)parametric survival models–such as the Cox proportional hazards model [13] and related treatment-effect models [58]–our model does not require covariate effects on the hazard to be constant over time.

Instead, we adopt NN-based discrete-time survival models that estimate conditional hazards through a series of binary classification tasks [10, 24], following their adaptation for HTE estimation with binary treatments by Curth et al. [14]. From the estimated conditional hazards, we can derive more practically relevant measures, such as treatment-specific survival functions and the (restricted) mean survival time, which in turn allow us to define treatment effects and dose-response relationships. The core component of DoseSurv is the radial basis function (RBF) hazard estimator, which models the survival hazard as a treatment-specific continuous function of dosage via Gaussian RBFs, leveraging the varying-coefficient network framework introduced by Nie et al. [51]. Through shared representations, DoseSurv also draws on information across different treatment options. Furthermore, it incorporates balanced representation learning via integral probability metric (IPM) regularization to mitigate various covariate shifts encountered in TTE and observational datasets [62, 14]. Here, event occurrence (survival), censoring (e.g., loss to follow-up), as well as treatment and dosage assignment, typically depend on individual patient characteristics leading to different covariate shifts [14]. We therefore evaluate DoseSurv on real and synthetic datasets spanning these covariate shifts and different treatment scenarios to assess its ability to mitigate the resulting biases.

**Contributions**   Our contributions are fivefold: (i) We address the problem of HTE estimation for single- and multi-arm treatment regimens with mutually exclusive continuous-valued (dosage-dependent) and non-dosage treatment options, as well as TTE outcomes–a setting that has been largely overlooked despite its high practical relevance. (ii) We propose DoseSurv, a novel varying-coefficient network based on layer-specific RBFs and designed for HTE estimation in settings such as the one described above. (iii) We adopt a computationally efficient IPM regularization technique to mitigate various biases arising in HTE estimation from observational TTE data and continuous-valued treatments. (iv) Moreover, we create synthetic TTE datasets for different continuous-valued treatment types, and (v) we propose an extension of the classical *Twins* dataset, as a real-world benchmark for continuous-valued treatment settings with TTE data. Our results show that DoseSurv outperforms existing baselines under different treatment scenarios and covariate shifts.

## 2   Related Work

ML methods for estimating HTEs from binary interventions are well-studied and include model-agnostic meta-learners (e.g., S- and T-learners) [40, 47, 72] and model-specific approaches like multi-task architectures that leverage shared information across treatment groups while also allowing treatment-specific modeling [62, 16, 2]. Implementations typically span tree-based models [69, 6] and NNs [34, 62, 63, 16], and most works consider binary interventions only.

For continuous-valued interventions, classical approaches use the generalized propensity score [30], while recent NN-based architectures directly model dose-response relationships. Some methods discretize dosage into bins with treatment-specific heads [60], offering regional flexibility at the expense of strict continuity; others adopt varying-coefficient formulations in which parameters change smoothly with dosage [51, 61], preserving continuity. Adversarial approaches have also been explored for continuous-valued interventions [9]; in TTE settings, however, censoring and temporal dependence complicate defining faithful discrimination targets and stable training objectives.

In survival analysis, traditional models such as the Cox proportional hazards model [13] have been extended using NNs [20, 37]. Non-parametric alternatives include random survival forests [32] and discrete-time NN approaches [10, 45, 24], alongside survival clustering methods that group samples with similar risk profiles [33]. Integrating HTE estimation directly into survival models remains comparatively underexplored: most existing work addresses binary treatments using tree-based [29] or NN approaches [14, 58], with no support for continuous-valued treatments.

We provide an extended review of methods for survival analysis and HTE estimation–including those targeting continuous-valued interventions–in Appendix D, and summarize the applicability of related neural approaches across settings in Table D.1.

## 3 Problem Statement

### 3.1 Definitions

We aim to analyze an observational TTE dataset $\mathcal{D}$ comprising $N$ individuals, each of whom is treated with one of $M$ different continuous-valued treatments. Throughout, we largely follow notation and discrete-time survival formulations similar to those in [14]. The dataset can then be represented as $\mathcal{D} = \{\boldsymbol{x}_i, \tilde{\tau}_i, \delta_i, (a_i, q_i)\}_{i=1}^N$, where each patient $i$ is a realization of random variables $(\boldsymbol{X}, \tilde{T}, \Delta, (A, Q)) \sim \mathbb{P}$. In this context, $\boldsymbol{X} \in \mathcal{X}$ refers to a vector of random variables associated with the patient covariates. Furthermore, we use $T \in \mathcal{T}$ and $C \in \mathcal{T}$ to represent the random variables for the event time (e.g., the time until death or onset of disease) and the censoring time (i.e., the time until loss of follow-up). The observed patient outcomes are given by $\Delta = \mathbb{1}(T \leq C)$, indicating whether either the event or censoring was observed, and $\tilde{T} = \min(T, C)$, representing the time of that observation. We denote the random variable for the treatment assignment as $A \in \mathcal{A} = \{0, \dots, M-1\}$. Each of these treatments $A \in \mathcal{A}$ is associated with a continuous random variable $Q \in \mathcal{Q} = [0, 1] \subset \mathbb{R}$, representing the (normalized) dosage for the respective treatment.[1] We treat time as discrete, with a finite time horizon $t_{\mathrm{m}}$, such that $\mathcal{T} = \{1, \cdots, t_{\mathrm{m}}\}$. Based on the notation above, the probability of the event occurring at time $\tau$ for a given patient with covariates $\boldsymbol{x}$ and intervention[2] $(a, q)$ who has not experienced an event prior to $\tau$ is defined as [14]

$$\lambda(\tau|\boldsymbol{x}, a, q) = \mathbb{P}(\tilde{T} = \tau, \Delta = 1 | \boldsymbol{X} = \boldsymbol{x}, A = a, Q = q, \tilde{T} \geq \tau), \tag{1}$$

which is also known as *conditional hazard*. From this, we can compute the survival function $S(\tau|\boldsymbol{x}, a, q) = \prod_{t \leq \tau}(1 - \lambda(t|\boldsymbol{x}, a, q))$, representing the probability that the event will not occur up until time $\tau$. To infer the conditional hazards (and thus the survival function), we follow a typical approach in discrete-time survival analysis by estimating $\lambda(\tau|\boldsymbol{x}, a, q)$ from covariates and time intervals using a non-parametric ML model. We transform the TTE data into *person-period* (long) format [66] and treat the estimation of the discrete conditional hazard function as a series of binary classification problems associated with different time intervals. More specifically, we can fit an NN model to the binary event indicator at each discrete time interval, with $\boldsymbol{x}$, $a$, $q$, and $\tau$ as predictors.

### 3.2 Relationship to Potential Outcomes Framework

In HTE estimation, our aim is not primarily to estimate outcomes conditional on the intervention, as presented above. Instead, we want to estimate the *potential outcome* that will occur in the event that a particular action is performed. Following the Rubin causal model [31], we can adopt the survival function as a potential outcome of interest under a potential intervention $(a, q)$ [14]

$$S_{a,q}(\tau|\boldsymbol{x}) = \mathbb{P}(T > \tau | \boldsymbol{X} = \boldsymbol{x}, \mathrm{do}(A = a, Q = q, C \geq \tau)), \tag{2}$$

where the do-operator [53] shows that a specific action is being taken. Here, we have to treat censoring as an action we can intervene on by setting $C \geq \tau$, so that potential outcomes reflect uncensored event times and causal effects are identifiable [64].

The HTE between two potential interventions $(a, q)$ and $(\tilde{a}, \tilde{q})$ may then, for example, be defined as the (time-dependent) difference in survival probabilities $\nu_{\mathrm{surv}}(\tau|\boldsymbol{x}) = S_{a,q}(\tau|\boldsymbol{x}) - S_{\tilde{a},\tilde{q}}(\tau|\boldsymbol{x})$. Alternatively, we may compute the restricted mean survival time (RMST) until the time horizon $t_{\mathrm{m}}$ under an intervention $(a, q)$ as a time-independent potential outcome of interest: $t_{a,q}^{\mathrm{rmst}}(\boldsymbol{x}) = \sum_{t=0}^{t_{\mathrm{m}}-1} S_{a,q}(t|\boldsymbol{x})$. The HTE can then be defined as $\nu_{\mathrm{rmst}}(\boldsymbol{x}) = t_{a,q}^{\mathrm{rmst}}(\boldsymbol{x}) - t_{\tilde{a},\tilde{q}}^{\mathrm{rmst}}(\boldsymbol{x})$ [14]. Moreover, for a fixed treatment $a$, the function $q \mapsto t_{a,q}^{\mathrm{rmst}}(\boldsymbol{x})$ defines a patient-specific dose-response curve that summarizes expected survival as a function of dosage.

To identify and quantify HTEs and dose-responses from TTE data, we make the following *assumptions*, which are standard in the literature on (dosage-dependent) treatment effect estimation [9, 60, 14] (Assumptions 1–3) and TTE outcomes [19, 50, 14] (Assumptions 4–5), and extend them for continuous-valued treatments. We denote the potential event time under a potential intervention $(a, q)$ as $T_{a,q}$.

---

[1] For simplicity, we focus here on a set of continuous-valued treatments only, i.e., $\mathcal{A} = \mathcal{A}^{\mathrm{cont}}$. However, DoseSurv can also accommodate the more general case with continuous-valued (dosage-dependent) and categorical (non-dosage) treatment options, i.e., $\mathcal{A} = \mathcal{A}^{\mathrm{cont}} \cup \mathcal{A}^{\mathrm{cat}}$; for the latter, $q$ is a dummy with no effect.

[2] In the following, we refer to the tuple $(a, q)$ as "intervention" to distinguish it from a "treatment" $a$ with associated "dosage" $q$.

**Assumption 1** (*Unconfoundedness*) For all $a \in \mathcal{A}$ and $q \in \mathcal{Q}$, the intervention and the potential event time are conditionally independent given covariates: $(A, Q) \perp T_{a,q} \mid \boldsymbol{X}$.

**Assumption 2** (*Positivity of Interventions*) For all $\boldsymbol{x} \in \mathcal{X}$ and all $a \in \mathcal{A}$:

(2a) *Positivity of Treatment*: There exists $\epsilon > 0$ such that $\epsilon < \mathbb{P}(A = a \mid \boldsymbol{X} = \boldsymbol{x}) < 1 - \epsilon$.

(2b) *Positivity of Dosage*: For any admissible dose $q \in \mathcal{Q}$ and any small interval $\mathcal{U} \subset \mathcal{Q}$ around $q$, $\mathbb{P}(Q \in \mathcal{U} \mid A = a, \boldsymbol{X} = \boldsymbol{x}) > 0$.

**Assumption 3** (*Consistency*) The potential event time will actually be observed under the observed intervention without being affected by any other external factors, i.e., if $A = a$ and $Q = q$, then $T = T_{a,q}$.

**Assumption 4** (*Independent Censoring*) For all $a \in \mathcal{A}$, $q \in \mathcal{Q}$, the potential event time and the censoring time are conditionally independent given covariates and intervention: $T_{a,q} \perp C \mid \boldsymbol{X}, (A, Q)$.

**Assumption 5** (*Positivity of Events and Censoring*) For all $\boldsymbol{x} \in \mathcal{X}$, $a \in \mathcal{A}$, $q \in \mathcal{Q}$, and all $t \in \mathcal{T}$:

(5a) *Positivity of Events*: $\mathbb{P}(T > t \mid \boldsymbol{X} = \boldsymbol{x}, A = a, Q = q) \geq \epsilon$ for some $\epsilon > 0$.

(5b) *Positivity of Censoring*: $\mathbb{P}(C > t \mid \boldsymbol{X} = \boldsymbol{x}, A = a, Q = q) \geq \epsilon$ for some $\epsilon > 0$.

Under Assumptions 1, 3, and 4, we can identify the interventional hazard from the observational one, i.e., $\lambda(\tau|\boldsymbol{x}, a, q) = \lambda_{a,q}(\tau|\boldsymbol{x}) = \mathbb{P}(T = \tau \mid T \geq \tau, \boldsymbol{X} = \boldsymbol{x}, \mathrm{do}(A = a, Q = q, C \geq \tau))$. The corresponding potential outcome survival function can then be computed as $S_{a,q}(\tau|\boldsymbol{x}) = \prod_{t \leq \tau} \left(1 - \lambda_{a,q}(t|\boldsymbol{x})\right)$. A brief mathematical derivation of this identification is given in Appendix F. Assumptions 2 and 5 further provide the overlap and non-degeneracy conditions for treatment, dosage, and at-risk sets required for non-parametric estimation [14]. We provide the causal diagram underlying the assumed data-generating process in Appendix G.

## 4 DoseSurv

### 4.1 Model Architecture

Based on the definition of the problem presented above, we propose an NN-based discrete-time survival model, called DoseSurv, which is depicted in Fig. 1, and is tailored for survival predictions in single- and multi-arm treatment regimens with dosage-dependent (and non-dosage) treatments. DoseSurv consists of a *representation network*, $\Phi : \mathcal{X} \to \mathcal{R}$, which is implemented as a fully connected NN and shared across all treatments $a$, dosages $q$, and times $\tau$. It is followed by treatment-specific *hazard estimators*, implemented as individual networks, each performing multi-output binary classification $h^{(a)} : \mathcal{R} \times \mathcal{Q} \to [0, 1]^{t_\mathrm{m}}$, which allow us to estimate the survival hazards at each time point $\tau$ for a specific continuous-valued treatment $a$

$$\hat{\lambda}(\tau|\boldsymbol{x}, a, q) = h_\tau^{(a)}(\Phi(\boldsymbol{x}), q), \tag{3}$$

where $h_\tau^{(a)}$ is the output of $h^{(a)}$ corresponding to time $\tau$. The treatment-specific estimators enable the learning of features unique to each respective treatment option. To enable the learning of the dosage dependency in the hazard estimates, we take two steps: (1) we add $q$ as an additional feature to the shared representation $\Phi(\boldsymbol{x})$. (2) more importantly, we use Gaussian RBFs to explicitly model the network parameters (weights and biases) as continuous functions of $q$, as described below.[3]

### 4.2 The RBF Hazard Estimator

For continuous-valued treatments $a$, we want to preserve the prominent role of $q$ and prevent its influence from vanishing in a high-dimensional feature space. Therefore, we implement the treatment-specific hazard estimators as flexible varying-coefficient networks, where the network parameters $\theta$ are functions of dosage $q$, following prior work by Nie et al. [51]. We depart from that formulation by modeling the network parameters as linear combinations of treatment- and layer-specific RBFs with learnable centers and bandwidths. Concretely, let the hazard estimator

---

[3] For non-dosage treatments $a \in \mathcal{A}^\mathrm{cat}$, $h^{(a)} : \mathcal{R} \to [0, 1]^{t_\mathrm{m}}$ is implemented as a standard fully connected NN.

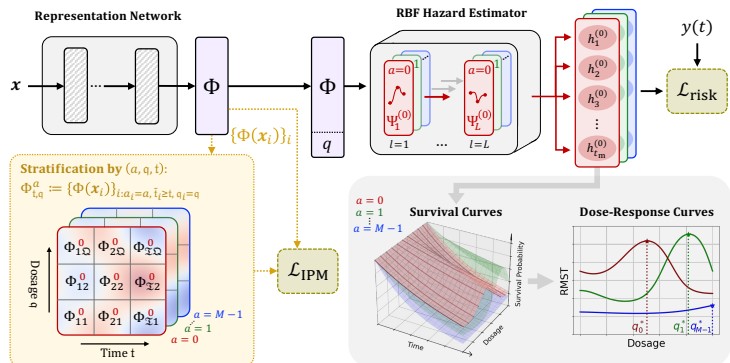

Figure 1: DoseSurv features a shared representation network that captures common features across treatments, dosages, and time intervals. To address covariate shifts arising from treatment-selection bias, dosage-selection bias, censoring, and event-induced shifts, we introduce an IPM regularization term based on a custom discretization of dosage ($q \to \mathfrak{q}$) and time ($t \to \mathfrak{t}$), enabling the learning of balanced representations $\Phi$ across treatments, dosages, and event times. For continuous treatments, $a \in \mathcal{A}^{\text{cont}}$, the model extends to the RBF hazard estimator, which consists of treatment-specific network heads. These heads use treatment- and layer-specific RBF bases $\{\psi_{l,k}^{(a)}\}_{k=1}^{K}$ to model parameters as continuous functions of $q$, ensuring continuity of discrete-time hazard estimates with respect to dosage. The hazard estimates allow for the computation of treatment- and dosage-specific survival curves, enabling dose-response analysis.

$h^{(a)}$ comprise $J_l$ parameters (weights and biases) in each layer $l \in \{1, \ldots, L\}$, which are given by $\boldsymbol{\theta}_l^{(a)}(q) = \left(\theta_{1,l}^{(a)}(q), \theta_{2,l}^{(a)}(q), \ldots, \theta_{J_l,l}^{(a)}(q)\right)^{\top}$. Each of these parameters, indexed by $j$, is modeled as a continuous function of $q$

$$\theta_{j,l}^{(a)}(q) = \sum_{k=1}^{K} \beta_{j,l,k}^{(a)} \psi_{l,k}^{(a)}(q), \tag{4}$$

where $\{\psi_{l,k}^{(a)}\}_{k=1}^{K}$ are the $K$ basis functions in layer $l$ for treatment $a$, and $\beta_{j,l,k}^{(a)}$ are the learnable coefficients [51]. We can therefore write

$$\boldsymbol{\theta}_l^{(a)}(q) = \boldsymbol{B}_l^{(a)} \boldsymbol{\Psi}_l^{(a)}(q) \tag{5}$$

with

$$\boldsymbol{B}_l^{(a)} = \begin{bmatrix} \beta_{1,l,1}^{(a)} & \cdots & \beta_{1,l,K}^{(a)} \\ \vdots & \ddots & \vdots \\ \beta_{J_l,l,1}^{(a)} & \cdots & \beta_{J_l,l,K}^{(a)} \end{bmatrix} \in \mathbb{R}^{J_l \times K}, \quad \boldsymbol{\Psi}_l^{(a)}(q) = \left(\psi_{l,1}^{(a)}(q), \psi_{l,2}^{(a)}(q), \ldots, \psi_{l,K}^{(a)}(q)\right) \in \mathbb{R}^{K}.$$

We propose the use of Gaussian RBFs as basis functions

$$\psi_{l,k}^{(a)}(q) = w_{l,k}^{(a)} \exp\left[-\left(q - c_{l,k}^{(a)}\right)^2 / \left(2(\sigma_{l,k}^{(a)})^2\right)\right] \tag{6}$$

with learnable centers $(c_{l,1}^{(a)}, c_{l,2}^{(a)}, \ldots, c_{l,K}^{(a)})$, shape parameters $(\sigma_{l,1}^{(a)}, \sigma_{l,2}^{(a)}, \ldots, \sigma_{l,K}^{(a)})$ and additional weights $(w_{l,1}^{(a)}, w_{l,2}^{(a)}, \ldots, w_{l,K}^{(a)})$.[4] This flexibility, together with the universal approximation property of Gaussian RBF expansions in $q$, implies that–given sufficiently many centers and suitable bandwidths–our $q$-dependent parameterization can represent any continuous dose-response function on compact intervals [52]. Moreover, Gaussian RBFs provide global support over the dosage range, potentially aiding in low-overlap regions. Furthermore, because complex survival likelihoods and their gradients interact nonlinearly with basis functions, our approach aims to boost optimization by learning not only the varying coefficients but also the properties of the basis functions themselves.

---

[4]While not explored here, coupling these amplitude weights across clinically similar treatments would provide a simple route to induce soft cross-treatment information sharing between hazard estimators.

## 4.3 Loss Function and Representation Balancing

DoseSurv uses both empirical risk minimization and representation balancing to learn hazard estimates in the presence of biases in treatment and dosage selection, as well as censoring and event occurrence.

**Empirical Risk Loss**  We minimize the empirical risk through the following loss term [24, 41]

$$\mathcal{L}_{\text{risk}} = \frac{1}{N} \frac{1}{t_{\text{m}}} \sum_{t=1}^{t_{\text{m}}} \sum_{i:\tilde{\tau}_i \geq t} \text{BCE}\left(y_i(t), h_t^{(a_i)}(\Phi(\boldsymbol{x}_i), q_i)\right), \tag{7}$$

where $y_i(t) = \mathbb{1}\{\delta_i = 1 \wedge \tilde{\tau}_i = t\}$ denotes the observed event indicator for patient $i$ in person-period (long) format. In this formulation, $\mathcal{L}_{\text{risk}}$ is proportional to the negative discrete-time hazard log-likelihood–that is, the average binary cross-entropy (BCE) computed over at-risk rows only. Minimizing it corresponds to maximum-likelihood estimation of per-interval hazards, and censoring is handled via the at-risk conditioning $i:\tilde{\tau}_i \geq t$.

**Representation Balancing**  To mitigate the effects of confounding, as well as censoring and event-induced shifts, DoseSurv aims to learn a representation $\Phi(\boldsymbol{x})$ in which features are balanced across at-risk populations at different times under different treatments and dosages. For this purpose, we adapt an IPM regularization scheme for binary treatments [14], to minimize the Wasserstein distance [68] between feature distributions of individual time-, treatment-, and dosage-specific subpopulations and the overall baseline population. We discretize the continuous dosage $q$ into $\mathfrak{Q}$ equally-sized bins $\{[0, 1/\mathfrak{Q}), [1/\mathfrak{Q}, 2/\mathfrak{Q}), \cdots, [(\mathfrak{Q}-1)/\mathfrak{Q}, 1]\}$, indexed by $\mathfrak{q} \in \{1, 2, \cdots, \mathfrak{Q}\}$, and denote by $\mathfrak{q}_i$ the dosage-bin index for sample $i$. To prevent oversegmentation into overly small subpopulations–which can reduce computational efficiency, generalizability, and stability–we also discretize time $t$ into $\mathfrak{T}$ intervals of lower temporal resolution, indexed by $\mathfrak{t} \in \{1, 2, \cdots, \mathfrak{T}\}$. Let $\tilde{\mathfrak{t}}_i$ be the coarse time-bin index for sample $i$, obtained by mapping the observed time $\tilde{\tau}_i$ to one of these $\mathfrak{T}$ intervals. We then define the IPM regularization loss as the sum of Wasserstein distances between the representations of the total baseline population and that of each subpopulation

$$\mathcal{L}_{\text{IPM}} = \frac{1}{|\mathcal{A}|} \frac{1}{\mathfrak{Q}} \frac{1}{\mathfrak{T}} \sum_{a \in \mathcal{A}} \sum_{\mathfrak{q}=1}^{\mathfrak{Q}} \sum_{\mathfrak{t}=1}^{\mathfrak{T}} \text{Wass}(\{\Phi(\boldsymbol{x}_i)\}_i^N, \{\Phi(\boldsymbol{x}_i)\}_{i:a_i=a, \tilde{\mathfrak{t}}_i \geq \mathfrak{t}, \mathfrak{q}_i=\mathfrak{q}}). \tag{8}$$

Consequently, $\mathcal{L}_{\text{IPM}}$ penalizes, for each treatment $a$, time bin $\mathfrak{t}$, and dosage bin $\mathfrak{q}$, the discrepancy in $\Phi$ between the corresponding at-risk subpopulation and the overall baseline population.[5] In principle, one could complement or replace our IPM regularization with observation reweighting tailored to dynamic survival settings with continuous treatments. Such an approach would require estimating generalized propensity scores for treatment and dosage given covariates, as well as conditional at-risk (survival) probabilities over time. In practice, these quantities are difficult to estimate reliably–especially with continuous dosages and time-varying risk sets–and often yield extreme or unstable weights under limited overlap, which can inflate variance and destabilize training. To avoid these issues, we focus on IPM-based representation balancing.

We calculate the total loss function as the sum of the empirical loss and the IPM regularization term

$$\mathcal{L} = \mathcal{L}_{\text{risk}} + \gamma \mathcal{L}_{\text{IPM}}. \tag{9}$$

Here, $\gamma$ is a hyperparameter that controls the influence of the regularization term. More details on the implementation of DoseSurv are provided in Appendix E.

## 5 Experiments

### 5.1 Experimental Setup

Following previous work on HTE estimation with continuous-valued treatments [60, 9] and TTE outcomes [14, 58], we first evaluate the performance of our model on synthetic data, which provides access to counterfactual outcomes and ground-truth survival curves for individual patients, and is therefore essential for a thorough evaluation of causal effect estimation. In addition, we evaluate DoseSurv on real-world data, namely the *Twins* dataset.

---

[5]For non-dosage treatments $a \in \mathcal{A}^{\text{cat}}$, stratification of at-risk populations reduces to $(a, \mathfrak{t})$; we retain the same normalization to keep IPM loss contributions comparable across continuous-valued and non-dosage arms.

### 5.1.1 Synthetic Datasets

We consider three simulated treatment scenarios (S1–S3) and create synthetic TTE datasets with different underlying outcome generation processes for each of them. Patient covariates are correlated in each of the datasets. In S1, we evaluate model performance for a single continuous-valued treatment, i.e., $\mathcal{A}_{S1} = \{0\}$. In S2, we consider two continuous-valued treatment options $\mathcal{A}_{S2} = \{0, 1\}$. Finally, we investigate DoseSurv's performance in a setting with one non-dosage treatment option $\mathcal{A}_{S3}^{cat} = \{0\}$ and one continuous-valued treatment option $\mathcal{A}_{S3}^{cont} = \{1\}$. Extensive descriptions of the data generation process are provided in Appendix A.1. Our dosage assignment mechanism includes a parameter $\eta$ that controls the level of dosage selection bias in the observed policy. When $\eta = 0$, the dosage is sampled from a uniform distribution between $[0, 1]$. As $\eta$ increases, dosage selection becomes increasingly confounded. Throughout all experiments, we simulate dosage assignment for $\eta \in \{0, 1, 2, 3, 4\}$. In addition, we simulate a fixed treatment selection bias for scenarios S2 and S3, which comprise multiple treatment options, and employ an informed censoring process, which introduces an additional censoring bias over time.

### 5.1.2 Ablations on DoseSurv Design

To examine the impact of certain design elements of DoseSurv, we conduct ablation studies. We compare the standard DoseSurv model (5 RBFs with layer-specific, learnable centers initialized at $\{0, 0.25, 0.5, 0.75, 1\}$) with 5 different variants: The first ablation (A1) corresponds to the standard DoseSurv model, except with fixed (non-learnable) RBF centers. For the second variant (A2), we replace the RBF basis with a truncated polynomial (power) basis

$$\psi_{l,k}^{(a)}(q) = \begin{cases} q^k & \text{for } k = 0, \ldots, d \\ \max(q - c_{l,k-d}^{(a)}, 0)^d & \text{for } k = d+1, \ldots, d+\kappa \end{cases} \tag{10}$$

to construct splines of degree $d = 2$, with $\kappa = 5$ learnable knots, $c_{l,k-d}^{(a)}$, initialized at $\{0, 0.25, 0.5, 0.75, 0.95\}$. This results in a total of $K = 8$ basis functions, and thus higher complexity than the standard DoseSurv model. The third ablation (A3) corresponds to the standard DoseSurv model, but with only three basis functions (3 centers initialized at $\{0.2, 0.5, 0.8\}$). For ablation A4, we do not add $q$ as an additional feature to the shared representation $\Phi(\boldsymbol{x})$, incorporating it only implicitly by modeling the network parameters using the RBFs that depend on $q$. In the final ablation (A5), we employ the varying-coefficient network design proposed by Nie et al. [51] for non-TTE data. This design features a truncated polynomial basis of degree $d = 2$ with $\kappa = 2$ non-learnable knots at $\{1/3, 2/3\}$, and omits $q$ as an additional input feature. This results in $K = 5$ basis functions, yielding a complexity comparable to our standard DoseSurv model.

### 5.1.3 Real-world Dataset

Additionally, we evaluate DoseSurv on the Twins dataset, a well-established benchmark in the HTE literature for binary treatments. This dataset provides survival times for twins born in the United States between 1989 and 1991 [3]. The original use of this dataset focuses on binary treatment assignment–classifying the heavier twin as "treated" and the lighter twin as "control" [46, 70]. First, we keep this notion and validate DoseSurv in a binary treatment scenario, without dosages, similar to [14]. Second, we extend the scope of the dataset by replacing the binary treatment indicator with the actual birth weight, treating birth weight as a continuous "dosage". This reinterpretation allows the dataset to represent two potential outcomes for each sample exposed to two distinct dosages. Details on the dataset and the implemented censoring mechanism are provided in Appendix A.2.

### 5.1.4 Baselines

We benchmark DoseSurv against well-established and state-of-the-art ML models for TTE analysis:

- **DeepSurv** [37]: A continuous-time survival model that extends the Cox proportional hazards model by incorporating NNs while retaining the assumption of proportional hazards.
- **DeepHit** [45]: A discrete-time NN method modeling the event-time distribution, trained with composite likelihood and ranking losses; permits non-proportional hazards.
- **RSF (Random Survival Forest)** [32]: A non-parametric ML ensemble of decision trees (bootstrap sampling, log-rank splits) for right-censored survival data.

- **NSC (Neural Survival Clustering)** [33]: A method combining NNs and clustering to model TTE data by grouping samples with similar survival patterns and learning hazard functions per cluster.

By default, we implement all baselines as single models (S-learner) with treatment and dosage information as additional covariates. For scenarios S2 and S3 where multiple treatment options are available, we also compare against treatment-specific versions (T-learner) of the baseline models described above–trained separately per arm with the corresponding dosage included as an input feature–and refer to them as **DeepSurv-T**, **DeepHit-T**, **RSF-T** and **NSC-T**. Moreover, we consider a version of DoseSurv without IPM regularization, **DoseSurv (no IPM)**. For the binary treatment scenario on the Twins dataset, we additionally compare DoseSurv against existing binary HTE survival models, namely **SurvITE** [14] and **BITES** [58].

### 5.1.5 Performance Measures

For simulated data, we have access to the ground-truth individual dose-response relationships for every treatment $a$ and dosage $q \in [0,1]$. We can therefore quantify how closely the estimated patient-specific dose-response curves match the true underlying curves across dosages and treatments using the mean integrated squared error (MISE)[6] [60, 9], with RMST as the outcome of interest

$$\text{MISE} = \frac{1}{|\mathcal{A}|} \frac{1}{N} \sum_{a \in \mathcal{A}} \sum_{i=1}^{N} \int_{q=0}^{1} \left( \hat{t}_{a,q}^{\text{rmst}}(\boldsymbol{x}_i) - t_{a,q}^{\text{rmst}}(\boldsymbol{x}_i) \right)^2 \, \mathrm{d}q. \tag{11}$$

For continuous-valued treatments $a \in \mathcal{A}^{\text{cont}}$, we also evaluate the mean dosage policy error (DPE) [60], i.e., the mean squared error between the ground-truth outcome under the estimated optimal dosage $\hat{q}_{i,a}^* = \arg\max_{q \in [0,1]} \hat{t}_{a,q}^{\text{rmst}}(\boldsymbol{x}_i)$ and the true optimal dosage $q_{i,a}^* = \arg\max_{q \in [0,1]} t_{a,q}^{\text{rmst}}(\boldsymbol{x}_i)$, maximizing RMST. This is given by

$$\text{DPE} = \frac{1}{|\mathcal{A}^{\text{cont}}|} \frac{1}{N} \sum_{a \in \mathcal{A}^{\text{cont}}} \sum_{i=1}^{N} (t_{a,\hat{q}_{i,a}^*}^{\text{rmst}}(\boldsymbol{x}_i) - t_{a,q_{i,a}^*}^{\text{rmst}}(\boldsymbol{x}_i))^2. \tag{12}$$

The performance measures described above require knowledge of the outcome generation mechanism and are thus impractical for real-world applications. Therefore, we also provide common metrics in survival analysis based on observed (factual) outcomes: the time-dependent C-index ($C^{\text{td}}$), the integrated Brier score (IBS), and the integrated negative binomial log-likelihood (INBLL). Details on these metrics and corresponding model performance are provided in Appendix B.

In the Twins dataset adapted for continuous-valued treatments, for each sample $i$, we have access to the event times $T_0^{(i)} = T_{0,q_i^{(0)}}^{(i)}$ and $T_1^{(i)} = T_{0,q_i^{(1)}}^{(i)}$ under the lower dosage (birth weight) $q_i^{(0)}$, and the higher dosage $q_i^{(1)}$ of the same treatment $\mathcal{A} = \{0\}$, but not across the whole dosage range. Instead of computing MISE, we therefore provide the MSE between the observed and the estimated HTE

$$\text{MSE} = \frac{1}{N} \sum_{i}^{N} (\hat{\nu}_{\text{rmst},i} - \nu_{\text{rmst},i})^2, \tag{13}$$

where $\nu_{\text{rmst},i} = \min(T_{0,q_i^{(1)}}^{(i)}, t_{\text{m}}) - \min(T_{0,q_i^{(0)}}^{(i)}, t_{\text{m}})$ and $\hat{\nu}_{\text{rmst},i} = \hat{t}_{0,q_i^{(1)}}^{\text{rmst}}(\boldsymbol{x}_i) - \hat{t}_{0,q_i^{(0)}}^{\text{rmst}}(\boldsymbol{x}_i)$ are the observed and estimated HTEs in terms of the RMST. Analogously, we compute the MSE between estimated and observed HTEs for the non-dosage (binary) treatment case, $\mathcal{A} = \mathcal{A}^{\text{cat}} = \{0,1\}$, where $T_1^{(i)}$ represents the survival time of the heavier twin (treated) and $T_0^{(i)}$ of the lighter twin (control).

## 5.2 Results and Discussion

### 5.2.1 Main Results

Fig. 2 shows model performance across five runs for scenarios S1–S3 under varying levels of dosage selection bias ($\eta \in \{0,1,2,3,4\}$). In all experiments, either DoseSurv or its derivative without IPM regularization consistently achieved the best performance in terms of both MISE and DPE,

---

[6]For non-dosage treatments $a \in \mathcal{A}^{\text{cat}}$, $t_{a,q}^{\text{rmst}}(\boldsymbol{x})$ is constant over $q$, reducing MISE to a regular MSE between predicted and ground-truth RMST.

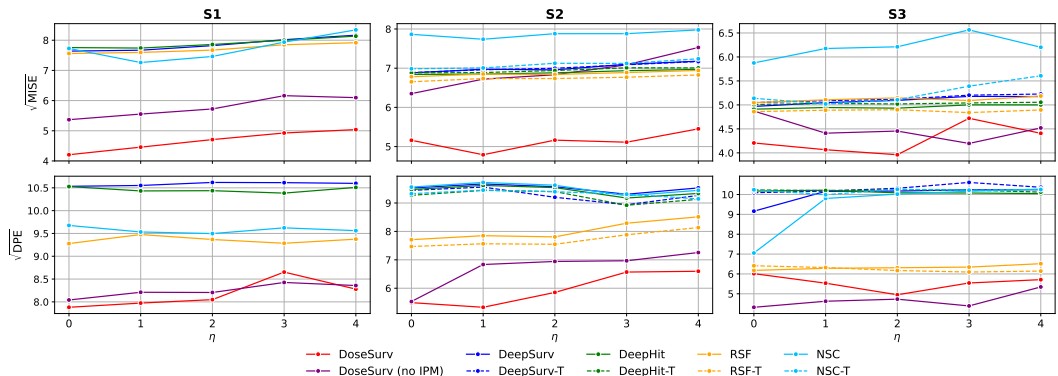

Figure 2: Model performance for scenarios S1–S3 under different levels of dosage-assignment bias, $\eta$, averaged across 5 runs. Smaller values for $\sqrt{\text{MISE}}$ and $\sqrt{\text{DPE}}$ indicate better performance.

outperforming all baseline models. As expected, performance generally declined slightly with increasing $\eta$ for both metrics. Notably, IPM regularization substantially improved performance in most scenarios, particularly in S2, which involves two continuous-valued treatments, and primarily in terms of MISE. IPM regularization also yielded performance improvements even in the absence of dosage-selection bias ($\eta = 0$) and treatment-selection bias (S1). We attribute this to its ability to mitigate other covariate shifts inherent in TTE data, such as event- or censoring-induced shifts.

The baseline models did not exhibit a clear performance trend. Among them, RSF and RSF-T generally achieved the lowest DPE. NSC performed comparably well in terms of DPE for S1, but showed the worst MISE performance in S2 and S3. DeepSurv(-T), DeepHit(-T), and NSC-T consistently demonstrated similar performance across all metrics and experiments.

Table 1 summarizes the performance on the real-world Twins dataset. Modeling the problem as a continuous treatment scenario and incorporating dosage information substantially improved performance across all models, highlighting the importance of continuous treatment modeling in survival analysis. DoseSurv demonstrated the best performance in estimating the HTE in both binary and continuous treatment settings, with substantial performance improvement observed in the continuous case. Among the baselines, DeepHit achieved the highest performance in the continuous setting, while RSF and DeepHit performed equally well in the binary setting.

Table 1: Model performance on the Twins dataset across 5 runs (mean±95% CI). For baselines, we also show the relative difference compared to DoseSurv.

| Methods | Twins (continuous) | Twins (binary) |
|---|---|---|
| | $\sqrt{\text{MSE}}$ | $\sqrt{\text{MSE}}$ |
| DeepSurv | 14.461±0.217 (+15.7%) | 16.235±0.151 (+4.1%) |
| DeepHit | 13.272±0.203 (+6.2%) | 15.974±0.090 (+2.4%) |
| RSF | 14.230±0.092 (+13.8%) | 15.923±0.022 (+2.1%) |
| NSC | 14.288±0.554 (+14.3%) | 16.694±0.123 (+7.0%) |
| BITES | —— | 16.177±0.161 (+3.7%) |
| SurvITE | —— | 19.599±4.725 (+25.6%) |
| DoseSurv | **12.501±0.063** | **15.601±0.531** |

### 5.2.2 Ablation Study Results

Figure 3 illustrates DoseSurv's performance for scenario S2 ($\eta = 1$), compared to various ablations (A1-A5), described in Section 5.1.2. The proposed design consistently achieved the best performance across all metrics compared to other variants. In particular, each individual design choice yielded a performance improvement over a DoseSurv implementation based on a traditional varying-coefficient network design. Moreover, the use of RBFs results in fewer basis functions and coefficients, $\beta_{j,l,k}^{(a)}$, and thus lower model complexity, for the same number of centers/knots.

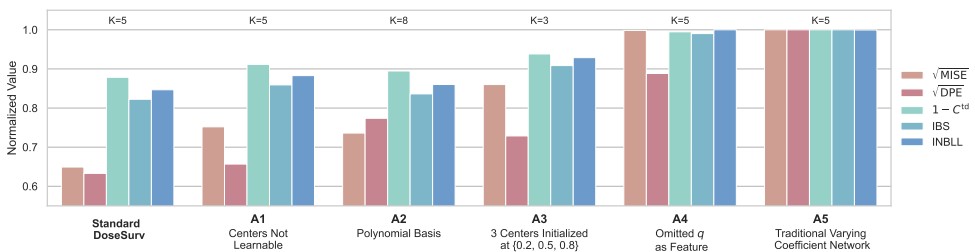

Figure 3: Impact of design choices (A1-A5) on the performance of DoseSurv in scenario S2 ($\eta=1$) averaged across 5 runs. All other hyperparameters were held constant. Performance metrics were normalized such that the highest mean value across all configurations equals 1. Lower values imply better performance. We indicate model complexity in terms of the number of basis functions, $K$.

## 6    Conclusion

In this paper, we presented an RBF-based varying-coefficient network for estimating HTEs from TTE data in single- and multi-arm treatment regimens with continuous-valued (dosage-dependent) and non-dosage treatment options. We demonstrated in various experiments the benefit of the varying-coefficient design in TTE settings with continuous treatments. Furthermore, we demonstrated significant performance improvements by incorporating an IPM regularization scheme tailored for these settings, effectively mitigating covariate shifts caused by confounding factors and the nature of TTE data.

**Limitations**    Like all HTE methods, our model relies on certain assumptions. In practice, leveraging expert and domain knowledge helps assess these and understand the impact of possible violations. Another challenge is tuning hyperparameters–especially the IPM regularization strength $\gamma$–under covariate shift. One way to address this is to select the largest $\gamma$ that preserves discrimination and calibration (e.g., via INBLL) on the validation set, balancing regularization and predictive power [14]. Finally, as with most deep learning models, performance benefits from sufficient training data; our sample size ablation in Appendix C.1 indicates degradation with small training sets.

**Broader Impact**    A promising future application of models like those explored in this work is adaptive clinical trials, where integrating observational TTE data and leveraging ML can optimize trial designs. This approach could enable dynamic adjustments in patient recruitment, treatment, and dosage options, ultimately improving trial outcomes and patient care. However, analyses must contend with covariate shifts–including treatment- and dosage-selection bias in observational settings and event- or censoring-induced shifts inherent to survival data–which, if unaddressed, can bias effect estimates and hinder optimal care.

## Acknowledgments

MG is supported by the EPSRC Centre for Doctoral Training in Health Data Science (EP/S02428X/1). CY is supported by a UKRI Turing AI Acceleration Fellowship (EP/V023233/1) and received additional funding through EPSRC grant EP/Y018192/1. TZ is supported by the Royal Academy of Engineering under the Research Fellowship scheme.

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

# Appendix

## A   Data Description

### A.1   Synthetic Data Generation

We generate covariates from a correlated multivariate normal distribution, $\boldsymbol{X} \sim \mathcal{N}(\boldsymbol{0}, \boldsymbol{\Sigma})$ with $\boldsymbol{\Sigma} = (1 - \rho)\mathbf{I} + \rho\mathbf{1}\mathbf{1}^\top$ and $\rho = 0.2$. We generate 20,000 samples with 40 covariates for scenario S1 and 20 covariates for S2 and S3. To simulate event times, we randomly sample a number of parameter vectors relevant to define the event-processes of each scenario, $\boldsymbol{u}_j^{\text{S1}}, \boldsymbol{u}_j^{\text{S2}}, \boldsymbol{u}_j^{\text{S3}} \sim \mathcal{N}(0, 1^2)$ and normalize them $\boldsymbol{v}_j^S = \boldsymbol{u}_j^S / \|\boldsymbol{u}_j^S\|$, where $\| \cdot \|$ is the Euclidean norm, and $S \in \{\text{S1, S2, S3}\}$. In scenarios S2 and S3, involving two treatment options, we aim to simulate an event process that reflects shared characteristics across treatments (e.g., common risk factors such as age) while also incorporating treatment-specific properties in the event process. We denote shared parameter vectors as $\boldsymbol{v}_j^S$ and treatment-specific parameter vectors as $\tilde{\boldsymbol{v}}_j^{S,a}$. Finally, for the three scenarios S1, S2, S3, we use the following event processes, where $\sigma(\cdot)$ is the sigmoid function:

---

**S1 (1 continuous-valued treatment option)**

$$\lambda_{0,q}(t|\boldsymbol{x}) = 0.25\sigma(\boldsymbol{v}_1^{\text{S1}}\boldsymbol{x})\sigma(-2\boldsymbol{v}_2^{\text{S1}}\boldsymbol{x}\sin(2\pi\boldsymbol{v}_3^{\text{S1}}\boldsymbol{x}(q+1)t/t_{\text{m}}) - $$
$$3.5\sin(6\pi q\sigma(\boldsymbol{v}_4^{\text{S1}}\boldsymbol{x} + 2/3) + \boldsymbol{v}_5^{\text{S1}}\boldsymbol{x}) - \text{sgn}(\boldsymbol{v}_6^{\text{S1}}\boldsymbol{x})q^2)$$

**S2 (2 continuous-valued treatment options)**

$$\lambda_{0,q}(t|\boldsymbol{x}) = \begin{cases} 0.1\sigma(\boldsymbol{v}_1^{\text{S2}}\boldsymbol{x}), \text{ if } t < 5 \\ 0.5\sigma(\boldsymbol{v}_1^{\text{S2}}\boldsymbol{x})\sigma\left(5\sin(5\pi q + \tilde{\boldsymbol{v}}_1^{\text{S2},0}\boldsymbol{x}) + 2\tilde{\boldsymbol{v}}_2^{\text{S2},0}\boldsymbol{x}q^2\right), \text{ if } t \geq 5 \end{cases}$$

$$\lambda_{1,q}(t|\boldsymbol{x}) = \begin{cases} 0.1\sigma(\boldsymbol{v}_1^{\text{S2}}\boldsymbol{x}), \text{ if } t < 5 \\ 0.5\sigma(\boldsymbol{v}_1^{\text{S2}}\boldsymbol{x})\sigma\left(5\,\text{sgn}(\tilde{\boldsymbol{v}}_1^{\text{S2},1}\boldsymbol{x})\sin(4\pi q + \tilde{\boldsymbol{v}}_2^{\text{S2},1}\boldsymbol{x}) + \tilde{\boldsymbol{v}}_1^{\text{S2},1}\boldsymbol{x}q^2\right), \text{ if } t \geq 5 \end{cases}$$

**S3 (1 non-dosage and 1 continuous-valued treatment option)**

$$\lambda_0(t|\boldsymbol{x}) = \begin{cases} 0.1\sigma(\boldsymbol{v}_1^{\text{S3}}\boldsymbol{x}), \text{ if } t < 5 \\ 0.2\sigma(3\tilde{\boldsymbol{v}}_1^{\text{S3},0}\boldsymbol{x} - 1), \text{ if } t \geq 5 \end{cases}$$

$$\lambda_{1,q}(t|\boldsymbol{x}) = \begin{cases} 0.1\sigma(\boldsymbol{v}_1^{\text{S3}}\boldsymbol{x}), \text{ if } t < 5 \\ 0.5\sigma(\boldsymbol{v}_1^{\text{S3}}\boldsymbol{x})\sigma\left(5\sin(5\pi q + \tilde{\boldsymbol{v}}_1^{\text{S3},1}\boldsymbol{x}) + 2\tilde{\boldsymbol{v}}_2^{\text{S3},1}\boldsymbol{x}q^2\right), \text{ if } t \geq 5 \end{cases}$$

---

Fig. A.1 (A) shows simulated ground-truth survival curves for scenario S1 for 6 different samples and dosages $q = 0.2, 0.5, 0.8$. Fig. A.1 (B) depicts the corresponding dose-response relationships, where we define the outcome of interest as the RMST. We define the covariate-dependent, informative, censoring process as

$$\lambda_{\text{cens}}^S(t|\boldsymbol{x}) = 0.02\sigma(\boldsymbol{v}_{\text{cens}}^S\boldsymbol{x}),$$

where $\boldsymbol{v}_{\text{cens}}^S$ is another randomly sampled and normalized parameter vector, introducing an additional censoring bias, and therefore covariate shift over time. Furthermore, we assume administrative censoring at $t_{\text{m}} = 30$. To assign dosages, we follow an approach similar to [9] and [61]. More specifically, for each continuous treatment option, $a$, we sample the dosages from a Beta distribution

$$q_a \sim \texttt{Beta}(\tilde{\eta}, b_{q_a^*})$$

with $b_{q_a^*} = (\tilde{\eta} - 1)/q_a^* + 2 - \tilde{\eta}$, where the mode of the Beta distribution is given by the true optimal dosage, $q_a^*$, maximizing the RMST for the respective patient under treatment $a$.[7] This mechanism mimics a realistic medical scenario in which the dosage assignment is informed by the true optimal

---

[7]To maintain symmetry, we sample $q_a \sim \texttt{Beta}(b_{q_a^*}, \tilde{\eta})$ with $b_{q_a^*} = (\tilde{\eta} - 1)/(1 - q_a^*) + 2 - \tilde{\eta}$ if $q_a^* < 0.5$.

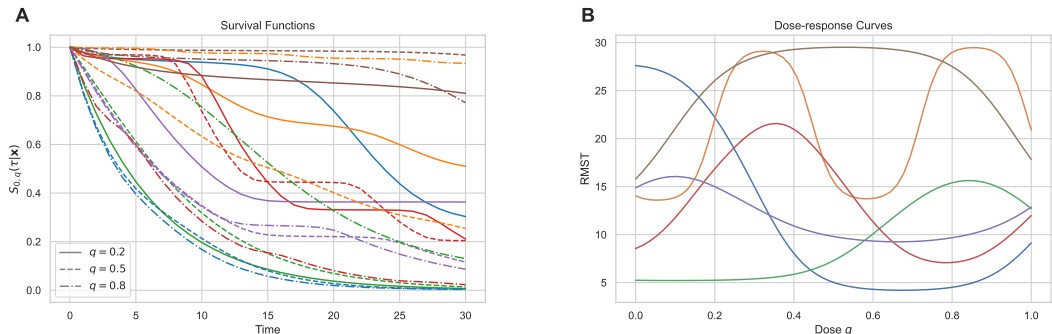

Figure A.1: Simulated (continuous) ground-truth survival curves for scenario S1 for 6 different samples and dosages $q = 0.2, 0.5, 0.8$ (A). Corresponding dose-response relationships with RMST as the outcome of interest (B).

dosage. The parameter $\eta = \tilde{\eta} - 1$ controls the dosage assignment, i.e., how much influence the covariates have on the dosage assignment. With $\eta = 0$, the dosage is sampled from a uniform distribution, while the dosage assignment becomes increasingly confounded towards the optimal dosage as $\eta$ increases.

To simulate confounded treatment assignment in scenarios S2 and S3, we design a mechanism that links the observed treatment type $a$ with the individual patient covariates. For this, we sample $M$ parameter vectors $\boldsymbol{u}_a \sim \mathcal{N}(0, 1^2)$ and normalize them $\boldsymbol{v}_a = \boldsymbol{u}_a/||\boldsymbol{u}_a||$. Then, we standardize the covariates $\boldsymbol{x}$ and compute the probabilities for observing treatment $a$

$$p_a = \frac{\exp(\boldsymbol{x}\boldsymbol{v}_a)}{\sum_{\alpha \in \mathcal{A}} \exp(\boldsymbol{x}\boldsymbol{v}_\alpha)}.$$

Next, we compute the modulated probabilities

$$\tilde{p}_a = \frac{(p_a)^\xi}{\sum_{\alpha \in \mathcal{A}}((p_\alpha)^\xi)},$$

where $\xi$ controls the treatment selection bias or confounding, i.e., how much influence the covariates have on the treatment decision (with $\xi = 0$ indicating a random treatment decision). For both scenarios S2 and S3 we choose $\xi = 1.5$ and sample the treatment from the multinomial distribution

$$a \sim \texttt{Multinomial}(\{\tilde{p}_\alpha\}_{\alpha \in \mathcal{A}}),$$

suitable for any number of treatment options. Figure A.2 illustrates the covariate shift in two selected covariates, $x_{11} + x_{19}$, between the two different treatment groups in scenario S2. The histograms represent the at-risk populations at different time points ($t = 0, 10, 20$) under different values of

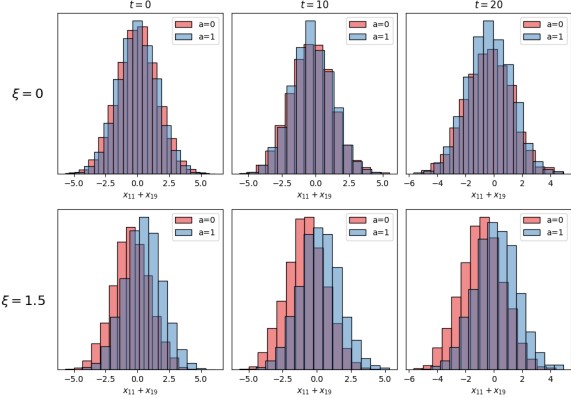

Figure A.2: Histograms of $x_{11} + x_{19}$ for at-risk populations across time for different treatments, $a = 0$ and $a = 1$, in scenario S2. Covariate shifts arise with increasing time $t$ and bias parameter $\xi$.

the treatment selection bias parameter ($\xi = 0$, 1.5). As $\xi$ increases, the covariate shifts become more pronounced. Additionally, shifts evolve over time due to differing event probabilities between treatment groups [14].

Across all scenarios, we employ 17,000 samples for training, 1,000 samples for validation, and 2,000 samples for testing.

## A.2 Twins Data

The Twins data is sourced from all US births between 1989–1991 [3], where we extract data on twins only, similar to [46, 70, 14]. Moreover, we select twins weighing less than 1,100 grams, extract 45 features predominantly related to parental factors, pregnancy, and birth, and include only samples with no missing features. We analyze 30-day survival, retaining twin pairs with no death after day 30; twins without a recorded death are treated as administratively censored at day 30. Moreover, we keep the original time resolution of 1 day as the length of the time intervals ($t_{\mathrm{m}} = 30$). We create a semi-synthetic observational TTE dataset by selecting one twin from each pair. For this, we use a similar mechanism as previously for the treatment assignment in the simulated scenarios S2 and S3: First, we randomly sample two parameter vectors $\boldsymbol{u}_0^{\mathrm{twin}}, \boldsymbol{u}_1^{\mathrm{twin}} \sim \mathcal{N}(0, 1^2)$ and normalize them $\boldsymbol{v}_0^{\mathrm{twin}} = \boldsymbol{u}_0^{\mathrm{twin}}/||\boldsymbol{u}_0^{\mathrm{twin}}||$, $\boldsymbol{v}_1^{\mathrm{twin}} = \boldsymbol{u}_1^{\mathrm{twin}}/||\boldsymbol{u}_1^{\mathrm{twin}}||$. After standardizing the covariates $\boldsymbol{x}$, we define the probability for observing the lower-weight twin or higher-weight twin as

$$p_a = \frac{\exp(\boldsymbol{x}\boldsymbol{v}_a^{\mathrm{twin}})}{\sum_{\alpha \in \{0,1\}} \exp(\boldsymbol{x}\boldsymbol{v}_\alpha^{\mathrm{twin}})}.$$

Next, we compute the modulated probabilities

$$\tilde{p}_a = \frac{(p_a)^\xi}{\sum_{\alpha \in \{0,1\}}((p_\alpha)^\xi)},$$

where we choose $\xi = 20$. We apply a synthetic censoring mechanism inspired by Schrod et al. [58] to introduce early censoring into samples which are not already administratively censored at day 30. As previously, we generate a normalized random vector $\boldsymbol{v}_{\mathrm{cens}}^{\mathrm{twin}}$ and compute a censoring score $s_i^{\mathrm{cens}}$ for each patient $i$, by computing the dot product with the standardized covariates: $s_i^{\mathrm{cens}} = \boldsymbol{v}_{\mathrm{cens}}^{\mathrm{twin}}\boldsymbol{x}_i$. After standardizing $s_i^{\mathrm{cens}}$ across all samples, we censor all remaining samples where $s_i^{\mathrm{cens}} > 0$ (i.e., $\sim 50\%$ of the remaining samples) at a random fraction $f_i \sim U(0, 1)$ of the sample's true observed survival time. Note that following this process we simulate a scenario where the independent censoring assumption is violated.

The final cohort contains 5,601 samples. For our analysis, we use 70% of samples for training. From the remaining samples, we extract 30% for validation and 70% for testing.

## B Metrics Based on Observed Outcomes

### B.1 Definitions

In the following, we provide definitions of the additional metrics used to measure the performance of survival models based on observed (factual) outcomes. Here we treat time as continuous and interpolate accordingly.

**Time-dependent C-index ($C^{\mathrm{td}}$):** Following the definition from [4, 43], we compute $C^{\mathrm{td}}$ as

$$C^{\mathrm{td}} = \mathbb{P}(\hat{S}(\tilde{\tau}_i|\boldsymbol{x}_i) < \hat{S}(\tilde{\tau}_i|\boldsymbol{x}_j)|\tilde{\tau}_i < \tilde{\tau}_j, \delta_i = 1).$$

This metric is *not* limited to settings where the proportional hazard assumption holds, but can be biased since only uncensored individuals are taken into account [49].

**Integrated (IPCW) Brier Score (IBS):** The IBS [26, 25] can be interpreted as a measure of the mean accuracy of the predicted survival probability over time, adjusting for censoring by incorporating inverse probability of censoring weights (IPCW). It is computed as

$$\mathrm{IBS} = \frac{1}{\max(\tilde{\tau}_i)} \int_0^{\max(\tilde{\tau}_i)} \frac{1}{N} \sum_{i=1}^{N} \left[ \frac{\hat{S}(t|\boldsymbol{x}_i)^2 \mathbb{1}\{\tilde{\tau}_i \leq t, \delta_i = 1\}}{\hat{G}(\tilde{\tau}_i)} + \frac{(1 - \hat{S}(t|\boldsymbol{x}_i))^2 \mathbb{1}\{\tilde{\tau}_i > t\}}{\hat{G}(t)} \right] dt,$$

where we denote the Kaplan-Meier estimator applied to the censoring times as $\hat{G}(t) = \mathbb{P}(C > t)$. Here, biases may arise due to dependencies between censoring distribution and covariates [42, 43].

**Integrated (IPCW) Negative Binomial Log-likelihood (INBLL):** The INBLL measures the performance of a survival model in terms of discrimination and calibration, averaged over time. It is calculated as [43]

$$\text{INBLL} = -\frac{1}{\max(\tilde{\tau}_i)} \int\limits_0^{\max(\tilde{\tau}_i)} \frac{1}{N} \sum_{i=1}^{N} \left[ \frac{\log[\hat{S}(t|\boldsymbol{x}_i)]\mathbb{1}\{\tilde{\tau}_i > t\}}{\hat{G}(t)} + \frac{\log[1 - \hat{S}(t|\boldsymbol{x}_i)]\mathbb{1}\{\tilde{\tau}_i \leq t, \delta_i = 1\}}{\hat{G}(\tilde{\tau}_i)} \right] dt.$$

We use this metric for hyperparameter optimization of the baseline models (see Section E.2).

## B.2 Results on Observable Metrics

Fig. B.1 illustrates the model performance, measured by $C^{\text{td}}$, IBS, and INBLL, for scenarios S1–S3, averaged over five runs. The results align closely with the synthetic data metrics presented in the main paper (see Fig. 2). Overall, DoseSurv consistently outperforms other models across most metrics and scenarios, with its variant without IPM regularization typically ranking second. Among the baseline models, DeepSurv generally achieves the best performance in terms of $C^{\text{td}}$, IBS, and INBLL in scenario S1, while RSF-T demonstrates high performance in scenarios S2 and S3.

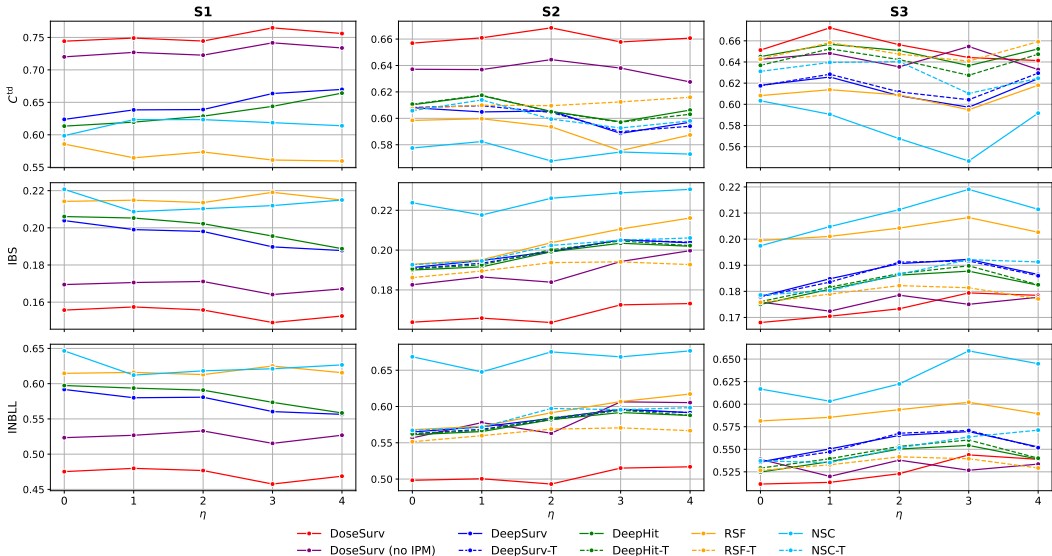

Figure B.1: Model performance for scenarios S1–S3, in terms of $C^{\text{td}}$, IBS and INBLL under different levels of dosage-assignment bias $\eta$, averaged across 5 runs. Greater values of $C^{\text{td}}$ and lower values of IBS and INBLL indicate better performance.

# C  Additional Experiments

## C.1 Ablation Study on Sample Size

We conducted an ablation study to examine the impact of training dataset size on model performance in scenario S1 ($\eta = 0$). We performed experiments using 5 different training sample sizes: 2,000, 6,000, 10,000, 14,000 and 18,000. Each training dataset was independently and randomly sampled from the combined training and validation pool. The same hyperparameters as in the main experiment for S1 were employed. For methods using early stopping (DoseSurv, DeepHit, DeepSurv, NSC), we reserved 10% of each sampled dataset for validation; RSF was trained on the full sampled set. The results are depicted in Fig. C.1. DoseSurv achieved the best performance across all sizes and metrics, except at the smallest size.

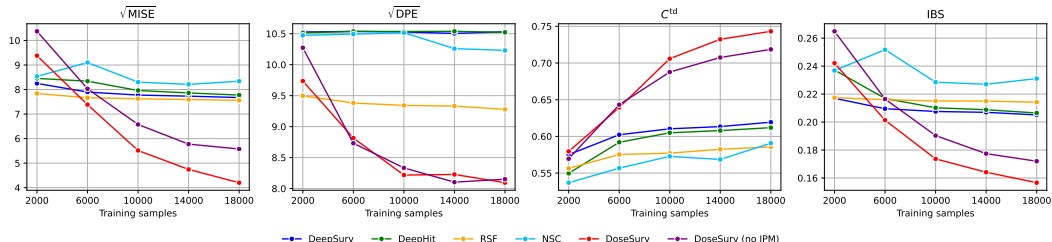

Figure C.1: Model performance across 5 runs (mean±95% CI) for scenario S1 ($\eta = 0$) for different training dataset sizes.

## C.2 Binary Treatment Scenario: Hormone Therapy in Breast Cancer Patients

We outline a potential application of DoseSurv to optimize binary treatment decisions, specifically hormone therapy for breast cancer patients. This external validation scenario has been previously explored by [57, 37, 58]. Although binary treatment settings are not the main focus of DoseSurv, the same approach could be applied to a continuous-valued treatment problem.

Following the methodology established in the referenced literature, we use data from 1,545 breast cancer patients in the Rotterdam Tumor Bank [22]. From this observational dataset, we extract six features: age, menopausal status, number of cancerous lymph nodes, tumor size, and progesterone and estrogen receptor statuses. Since treatment assignment in the Rotterdam dataset is non-random, we use these data for model training (85% training, 15% validation).

For testing, we use data from 686 patients enrolled in a randomized controlled trial conducted by the German Breast Cancer Study Group (GBSG) [59], ensuring external validation. The same six patient characteristics are extracted from this dataset. To simplify the analysis, survival times were discretized into 30 equally-sized intervals for DoseSurv and all baseline models.

The results, summarized in Table C.1, demonstrate that DoseSurv, although designed primarily for continuous-valued treatments, can be applied effectively in binary treatment settings. In particular, DoseSurv achieved better calibration accuracy (lower IBS and INBLL) than baseline models. Furthermore, IPM regularization led to improvements in model performance.

Table C.1: Model performance on the GBSG dataset across 5 runs (mean±95% CI).

| Methods | $C^{\text{td}}$ | IBS | INBLL |
|---|---|---|---|
| DeepSurv | 0.683±0.004 | 0.118±0.004 | 0.384±0.012 |
| DeepHit | 0.680±0.009 | 0.121±0.022 | 0.394±0.055 |
| RSF | 0.685±0.001 | 0.122±0.001 | 0.403±0.002 |
| NSC | 0.672±0.012 | 0.108±0.020 | 0.370±0.058 |
| BITES | 0.675±0.013 | 0.115±0.004 | 0.381±0.008 |
| SurvITE | 0.665±0.037 | 0.187±0.072 | 0.539±0.174 |
| DoseSurv (no IPM) | 0.670±0.008 | 0.110±0.005 | 0.356±0.012 |
| DoseSurv | **0.688±0.002** | **0.098±0.003** | **0.338±0.009** |

## C.3 Experiments under Different Time Discretization

Like other discrete-time NN survival models, DoseSurv supports flexible discretization of survival times into any number of intervals. In our main experiments, we used 30 time intervals, aligning with common practice in the literature [45, 14]. However, DoseSurv can readily operate under different temporal resolutions. Here, we perform additional experiments on data simulated using a finer time discretization of 60 intervals and compare DoseSurv to the continuous-time, CoxPH-inspired DeepSurv model, which may offer greater robustness. Table C.2 presents the corresponding results for data simulated under Scenario S1 ($\eta = 1$). We find that DoseSurv yields metrics comparable to those obtained with 30 intervals in the main experiment, whereas DeepSurv continues to underperform relative to DoseSurv.

Table C.2: Model performance of DoseSurv and DeepSurv under Scenario S1 ($\eta = 1$) with a finer time discretization (60 intervals). For reference, we include performance metrics achieved by DoseSurv in the corresponding main experiment with 30 intervals. Metrics are averaged over 5 runs.

| Intervals | Method | $\sqrt{\text{MISE}}$ | $\sqrt{\text{DPE}}$ | $C^{\text{td}}$ | IBS |
|---|---|---|---|---|---|
| 30 | DoseSurv | 4.46 | 7.97 | 0.75 | 0.16 |
| 60 | DoseSurv | 5.51 | 8.00 | 0.76 | 0.14 |
|  | DeepSurv (CoxPH) | 8.44 | 10.43 | 0.61 | 0.20 |

## C.4 Sensitivity Analysis

To assess the robustness of DoseSurv to hyperparameter choices, we conducted a sensitivity analysis under Scenario S1 ($\eta = 1$). We varied the learning rate (lr), batch size, and IPM regularization strength ($\gamma$), while keeping all other hyperparameters fixed. Results are averaged over 3 independent runs. The model demonstrates consistent performance across a broad range of settings, indicating relative stability with respect to these key hyperparameters.

Table C.3: Sensitivity of DoseSurv to learning rate, batch size, and IPM regularization under Scenario S1 ($\eta = 1$). Metrics are averaged over 3 runs.

| lr | Batch Size | $\gamma$ | $\sqrt{\text{MISE}}$ | $\sqrt{\text{DPE}}$ | $C^{\text{td}}$ | IBS |
|---|---|---|---|---|---|---|
| 0.001 | 500 | 0.01 | 4.58 | 7.94 | 0.75 | 0.16 |
| 0.001 | 500 | 0.10 | 4.48 | 7.88 | 0.75 | 0.16 |
| 0.001 | 1000 | 0.01 | 4.66 | 8.09 | 0.75 | 0.16 |
| 0.001 | 1000 | 0.10 | 4.70 | 8.07 | 0.75 | 0.16 |
| 0.005 | 500 | 0.01 | 5.47 | 8.39 | 0.73 | 0.17 |
| 0.005 | 500 | 0.10 | 5.73 | 8.43 | 0.72 | 0.17 |
| 0.005 | 1000 | 0.01 | 4.81 | 8.04 | 0.74 | 0.16 |
| 0.005 | 1000 | 0.10 | 4.95 | 8.10 | 0.74 | 0.16 |

## C.5 Visualizing Dose-Response Curves

Fig. C.2 shows an example dose-response curve predicted by DoseSurv in a single run under scenario S1 ($\eta = 0$) for a selected sample in red. The corresponding ground truth curve is depicted in green. In practice, dose-response curves can be obtained by evaluating the model for a range of dosage values $q$ and using the predicted RMST as the outcome of interest.

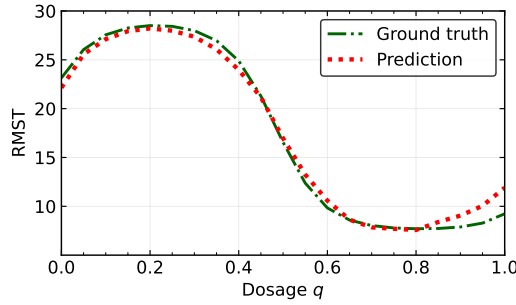

Figure C.2: Example of predicted and ground-truth dose-response curves under scenario S1 ($\eta = 0$).

# D  Extended Literature Review

## D.1  ML Methods for HTE Estimation.

Inferring HTEs of *binary* treatments has been studied extensively in the recent ML literature. Following Curth and van der Schaar [16], methods can typically be categorized into *model-specific* and *model-agnostic* approaches. The latter, also referred to as "meta-learners" [40], can be implemented using any kind of ML method. The most common types of meta-learners (*indirect estimators*) aim to obtain the HTE as the difference between regression estimates of the two potential outcome functions. T-learners [40, 47], fit two (or more) regression models (one for each treatment group) separately, while S-learners [23, 34, 40, 72] fit a single regression model where the treatment variable is added as an extra feature. Meta-learners have been implemented using a wide range of ML models, particularly tree-based models [23, 47, 69, 40, 6] and NNs [34, 72]. On the other hand, *model-specific* approaches rely on unique properties of the underlying ML model. These include approaches which could be described as *hybrids* between S- and T-learners [15]. Such methods are typically implemented as "multi-task learning" models, which learn different treatment responses simultaneously, allowing some information flow between treatments [1, 2, 62, 63, 16]. Beyond architecture, balancing and reweighting strategies are also used to mitigate confounding and covariate shift, including IPM-based representation balancing [62], counterfactual representation learning with balancing weights [5], importance-sampling reweighting [27], and propensity-based regularization [63].

HTE estimation of *continuous-valued* interventions is less explored in the literature. Traditional statistical approaches typically rely on the generalized propensity score [30]. The dose-response-networks proposed by Schwab et al. [60] extend the multi-task learning NN architecture for binary treatments, first presented by Shalit et al. [62], by subdividing the treatment-specific layers, into multiple network heads, each assigned to a different dosage interval. This facilitates the regression of the dose-response function across different dosage regions. A drawback of this architecture is that bin-specific heads produce separate outputs for different dosage intervals, breaking the continuity in the dose-response curves. Inspired by [28], Nie et al. [51] aim to overcome this limitation by employing a varying-coefficient network (VCNet) in the prediction head. Here, the network weights are modeled as continuous functions of the dosage. Schweisthal et al. [61] extended this approach to learn the effect of combinations of multiple dosage-dependent treatments. A different approach by Bica et al. [9] extends the GAN framework proposed by Yoon et al. [70], and introduced a hierarchical discriminator specifically tailored for continuous-valued treatment settings. However, leveraging adversarial learning for counterfactual survival analysis is challenging due to censoring and time structure, which complicate defining faithful discrimination targets and stable training objectives. Existing balancing methods for continuous-valued treatments assume clustered data [11], which limits its applicability in settings where natural clusters are absent. Moreover, Bellot et al. [8] provide a generalization bound, but their method relies on estimating complex integrals over representation and treatment spaces, which makes it likely to be sensitive to function and kernel choices.

## D.2  Survival Analysis and HTE Inference with TTE Data.

ML methods for survival analysis have gained significant attention in recent years. Notable approaches include adaptations of the Cox-proportional hazards model [13], where traditional linear predictors are replaced with feedforward NNs [20, 37]. In addition, TTE has been modeled via accelerated failure time (AFT) regression [35] and deep generative models [55]. Random survival forests have also emerged as a flexible approach to predict cumulative hazard functions in a non-parametric tree-ensemble manner [32]. Deep learning-based classifiers [10, 24, 56, 41] have gained popularity for this purpose. Alternatively, Lee et al. [45] parameterize the likelihood in discrete-time settings using the probability mass function. Finally, survival clustering methods have been proposed to identify groups of samples based on their survival patterns [7, 33].

Despite advances in survival analysis, adapting these models for HTE estimation remains understudied. Early investigations focus primarily on binary treatment settings, leveraging tree-based methods such as causal trees [71], Bayesian additive regression trees (BART) [29], and random forests [67] to measure the effect of binary treatments on expected survival time. More recent approaches, such as that by Chapfuwa et al. [12], explore representation learning and generative modeling to estimate event times for different treatments. A seminal work by Curth et al. [14] adopts a multi-task NN architecture with time- and treatment-specific network heads to predict discrete conditional hazard

functions, incorporating balanced representations to address covariate shifts. Schrod et al. [58] simplify this approach by combining a dichotomous multi-task architecture with a Cox-proportional hazard loss [37]. However, such works focus on binary interventions. In comparison, our DoseSurv model is more versatile and can handle single- and multi-arm treatment regimens with mutually exclusive continuous-valued (dosage-dependent) and non-dosage treatment options.

### D.3    Comparison of DoseSurv with Related Work

Table D.1 compares key ML methods for HTE estimation, specifically targeting continuous-valued treatments or TTE outcomes. Unlike related work, DoseSurv uniquely integrates TTE analysis with complex treatment settings that include continuous-valued treatments, non-dosage treatments, and mixed treatment scenarios. Due to the lack of HTE estimation models for such settings, our baseline comparisons rely on standard survival models, implemented as model-agnostic T- and S-learners. Although not explicitly studied here, DoseSurv is generally also applicable to treatment scenarios with more than two treatment options.

Table D.1: Overview of selected key neural methods for HTE estimation under continuous-valued treatments or TTE outcomes and their applicability across different settings.

| Model | Continuous-valued (Dosage) Treatments | Mixed Dosage and Non-dosage Options | More than Two Treatment Options | TTE Outcomes |
|---|---|---|---|---|
| DRNet [60] | ✓ | ✗ | ✓ | ✗ |
| SCIGAN [9] | ✓ | (✓) | ✓ | ✗ |
| VCNet [51] | ✓ | ✗ | ✗ | ✗ |
| SurvITE [14] | ✗ | ✗ | ✗ | ✓ |
| BITES [58] | ✗ | ✗ | ✗ | ✓ |
| **DoseSurv (ours)** | ✓ | ✓ | ✓ | ✓ |

# E    Implementation

## E.1    DoseSurv

The source code of DoseSurv will be made available at: https://github.com/mgoegl/DoseSurv.

Experiments were performed on a single GPU machine (specifications: CPU – Intel Xeon W5-2445, GPU – NVIDIA RTX A4000, and RAM – 64GB DDR5, OS – Linux). DoseSurv is implemented in PyTorch. The training time of DoseSurv was under 30 seconds per run for Scenario S1 ($\eta = 0$) without IPM regularization. Per-epoch training time generally ranges from under a second to a few seconds, depending on whether IPM regularization is applied. Overall, computation time increases with the number of IPM bins.

Across all main experiments, we implemented DoseSurv with 1 hidden layer in the representation network and 3 hidden layers in each treatment-specific head of the RBF hazard estimator, each layer comprising 100 nodes. For each layer in the RBF hazard estimator, we use 5 Gaussian RBFs with centers, initialized at $\{0, 0.25, 0.5, 0.75, 1\}$. The shape parameters $\sigma_{l,k}^{(a)}$ are initialized at 0.7. For each experiment, we chose $\gamma$ from $\{0.1, 0.01\}$ based on the lowest INBLL achieved on the validation data. For experiments on synthetic data, we optimized $\gamma$ under $\eta = 2$. Network parameters were optimized using Adam optimizer [38] with learning rate of 0.001. We use `ReLU` activation functions, dropout probability of 0.1, and a batch size of 500. Moreover, we implement DoseSurv with batch normalization layers, and employ early stopping after 30 epochs without model improvement on the validation data. We adopt the implementation of the finite-sample approximation of the Wasserstein distance from [14], which in turn follows the implementation in [5, 17], and improve computational efficiency by parallelizing its computation across treatments $a$, dosage bins $q$, and time bins $t$. We use the same parameters, i.e., an entropic regularization strength of $\lambda_{\text{Wass}} = 10$ and 10 Sinkhorn iterations. Moreover, we choose $\mathfrak{Q} = 3$ and $\mathfrak{T} = 5$.

Deviating from the primary experiments, we configure DoseSurv for the additional experiment described in C.2 with a representation network consisting of 3 layers of 200 nodes each and a

representation size of 50. The treatment-specific hazard estimators are designed with 2 layers of 100 nodes each, without applying batch normalization. Moreover, we choose $\mathfrak{Q} = 1$ and $\mathfrak{T} = 10$.

## E.2 Baseline Models

We benchmarked DoseSurv against publicly available implementations of four state-of-the-art machine and deep learning models for survival analysis, **DeepSurv**[8], **DeepHit**[8], **RSF**[9] and **NSC**[10]. We adopt these implementations as both T- and S-learners. First, we standardize covariates and treatment-specific dosages. For the standard (S-learner) baselines, we use slot-based (continuous one-hot) encoding of treatment and dosage information. We add the dosage information for each treatment option as additional continuous covariates. In scenarios with more than one treatment option, we set the dosage values for unobserved treatments to $-10$, which does not normally fall within the standardized dosage range, ensuring that the models do not mistakenly interpret this value as a valid dosage. For categorical (non-dosage) treatments, we set the dosage parameter for the observed treatment to 1. In addition, for scenarios S2 and S3 with more than one treatment group, we implement T-learner versions of the baseline models: **DeepSurv-T**, **DeepHit-T**, **RSF-T**, and **NSC-T**. Here, we split the data into the respective treatment groups, and train separate versions of the model on each treatment group individually, including the corresponding dosage as an additional covariate.

We optimize hyperparameters based on a grid search. We choose the optimal hyperparameter combination based on the minimum INBLL, which comprises both calibration and discriminative performance. For DeepSurv we optimize the number of layers and nodes, choosing from $\{[100], [100, 100], [100, 100, 100], [200, 200]\}$. Additionally, for DeepHit, we optimize the parameters $\alpha_{\text{DeepHit}} \in \{0.2, 0.5\}$ and $\sigma_{\text{DeepHit}} \in \{0.1, 1\}$, which control the contribution and properties of the ranking loss. For NSC, we choose network sizes of the mixture weights and survival networks from $\{[100, 100, 100], [200, 200]\}$. Furthermore, we choose the number of components for the mixture from $\{4, 6\}$, and the size of the latent cluster representation from $\{10, 100\}$. For RSF, we optimize the number of trees $\{100, 300\}$ as well as the minimum numbers of samples required to split an internal node ($\{6, 12\}$) or be at a leaf node ($\{3, 6\}$). For all network-based baseline models, as for DoseSurv, we use a dropout rate of 0.1, a batch size of 500, batch normalization, and early stopping after 30 epochs.

For experiments on the Twins and breast cancer datasets under binary treatments, we additionally compare against adapted public implementations of the HTE models **BITES**[11] and **SurvITE**[12] to ensure compatibility with our environment. For SurvITE, we choose the standard network architecture (representation network: $[100, 100, 100]$; hazard estimators: $[100, 100]$). For BITES, we choose the standard representation network of size $[7, 5]$, and optimize the treatment-specific heads from $\{[5, 3], [3]\}$. As for DoseSurv, we choose the IPM regularization strength for both models from $\{0.01, 0.1\}$ and BITES's blur parameter from $\{0.05, 0.1\}$.

## F  Remarks on HTE Estimation and Assumptions

In HTE inference, we generally aim to estimate

$$\nu(\boldsymbol{x}) = \mathbb{E}\left[Y_1 - Y_0 | \boldsymbol{X} = \boldsymbol{x}\right]$$
$$= \mathbb{E}\left[Y_1 | \boldsymbol{X} = \boldsymbol{x}\right] - \mathbb{E}\left[Y_0 | \boldsymbol{X} = \boldsymbol{x}\right]$$

where $Y_0$ and $Y_1$ are two continuous potential outcomes under treatments 0 and 1, and $\boldsymbol{X} = \boldsymbol{x}$ are the patient covariates. The potential outcomes can also be expressed using the do-operator [53] indicating an intervention to be made:

$$\nu(\boldsymbol{x}) = \mathbb{E}\left[Y | \text{do}(A = 1), \boldsymbol{X} = \boldsymbol{x}\right] - \mathbb{E}[Y | \text{do}(A = 0), \boldsymbol{X} = \boldsymbol{x}].$$

---

[8]https://github.com/havakv/pycox
[9]scikit-survival [54]
[10]https://github.com/Jeanselme/NeuralSurvivalClustering
[11]https://github.com/sschrod/BITES
[12]https://github.com/chl8856/survITE

For HTE estimation between two dosage-dependent interventions $(a, q)$ and $(\tilde{a}, \tilde{q})$ and the time-specific survival probabilities as outcome of interest, this translates into

$$\nu_{\text{surv}}(t|\boldsymbol{x}) = \mathbb{P}(T_{a,q} > t|\boldsymbol{X} = \boldsymbol{x}, \text{do}(C \geq t)) -$$
$$\mathbb{P}(T_{\tilde{a},\tilde{q}} > t|\boldsymbol{X} = \boldsymbol{x}, \text{do}(C \geq t))$$
$$= S_{a,q}(t|\boldsymbol{x}) - S_{\tilde{a},\tilde{q}}(t|\boldsymbol{x}).$$

However, since ground-truth survival functions are unknown in real-world settings, we focus on estimating the HTE in terms of the RMST (up until the time horizon, $t_{\text{m}}$) given by

$$\nu_{\text{rmst}}(\boldsymbol{x}) = \mathbb{E}\left[\min(T_{a,q}, t_{\text{m}})|\boldsymbol{X} = \boldsymbol{x}\right] - \mathbb{E}\left[\min(T_{\tilde{a},\tilde{q}}, t_{\text{m}})|\boldsymbol{X} = \boldsymbol{x}\right]$$
$$= t_{a,q}^{\text{rmst}}(\boldsymbol{x}) - t_{\tilde{a},\tilde{q}}^{\text{rmst}}(\boldsymbol{x})$$
$$= \sum_{t=0}^{t_{\text{m}}-1} S_{a,q}(t|\boldsymbol{x}) - \sum_{t=0}^{t_{\text{m}}-1} S_{\tilde{a},\tilde{q}}(t|\boldsymbol{x})$$
$$= \sum_{t=0}^{t_{\text{m}}-1} \nu_{\text{surv}}(t|\boldsymbol{x}).$$

Since $S_{a,q}(t|\boldsymbol{x}) = \prod_{\tau \leq t}(1 - \lambda_{a,q}(\tau|\boldsymbol{x}))$, we therefore need to compute $\lambda_{a,q}(\tau|\boldsymbol{x})$ in order to find unbiased estimates for both $\nu_{\text{surv}}(t|\boldsymbol{x})$ and $\nu_{\text{rmst}}(\boldsymbol{x})$. We can rewrite

$$\lambda_{a,q}(\tau|\boldsymbol{x}) =$$
$$= \mathbb{P}(T = \tau | T \geq \tau, \boldsymbol{X} = \boldsymbol{x}, \text{do}(A = a, Q = q, C \geq \tau))$$
$$= \mathbb{P}(T_{a,q} = \tau | T_{a,q} \geq \tau, \boldsymbol{X} = \boldsymbol{x}, \text{do}(C \geq \tau)).$$

Using Assumptions 1, 3–4, outlined in Section 3.2, we obtain [14]

$$\lambda_{a,q}(\tau|\boldsymbol{x}) =$$
$$\stackrel{(1)}{=} \mathbb{P}(T_{a,q} = \tau | T_{a,q} \geq \tau, \boldsymbol{X} = \boldsymbol{x}, A = a, Q = q, \text{do}(C \geq \tau))$$
$$\stackrel{(4)}{=} \mathbb{P}(T_{a,q} = \tau | T_{a,q} \geq \tau, \boldsymbol{X} = \boldsymbol{x}, A = a, Q = q, C \geq \tau)$$
$$\stackrel{(3)}{=} \mathbb{P}(T = \tau | T \geq \tau, \boldsymbol{X} = \boldsymbol{x}, A = a, Q = q, C \geq \tau)$$
$$= \mathbb{P}(\tilde{T} = \tau, \Delta = 1 | \boldsymbol{X} = \boldsymbol{x}, A = a, Q = q, \tilde{T} \geq \tau)$$
$$= \lambda(\tau|\boldsymbol{x}, a, q),$$

which can be estimated non-parametrically under the additional assumptions that treatment and dosage assignment (Assumption 2) and the survival and censoring processes (Assumption 5) satisfy positivity (i.e., are non-deterministic).

# G  Causal Diagram

The directed acyclic graph (DAG) [53] for our setting is shown in Fig. G.1 and follows prior work on HTE estimation with TTE data [14, 65], extended here to include the dosage $Q$. We define $\mathcal{N}(t) = \mathbb{1}(\tilde{T} \leq t, \Delta = 1)$ and $\mathcal{N}_C(t) = \mathbb{1}(\tilde{T} \leq t, \Delta = 0)$ as counting processes indicating, over time, whether an event (death) or, respectively, censoring has occurred. By definition, $\mathcal{N}(t)$ and $\mathcal{N}_C(t)$ are 0 for $t = 0$ and switch to 1 for $t \geq \tilde{T}$, i.e., when either the event or censoring occurs at time $\tilde{T}$.

$\boldsymbol{X}$ consists of (possibly overlapping) subgroups $\boldsymbol{X}_1$, $\boldsymbol{X}_2$, $\boldsymbol{X}_3$, and $\boldsymbol{X}_4$ of covariates that influence (1) the event process, (2) treatment assignment, (3) dosage assignment, and (4) the censoring process. These subgroups and their causal influences are associated with covariate shifts arising from (1) event-induced bias, (2) treatment-selection bias, (3) dosage-selection bias, and (4) censoring bias, respectively [14].

The causal diagram describes how $\mathcal{N}$ and $\mathcal{N}_C$ evolve over time under the influence of covariates $\boldsymbol{X}$, treatment $A$, and dosage $Q$, and is in line with the Assumptions formulated in Section 3.2. The

temporal dependencies shown ($\mathcal{N}(t-1) \to \mathcal{N}(t)$ and $\mathcal{N}_C(t-1) \to \mathcal{N}(t), \mathcal{N}_C(t)$) reflect the fact that these counting processes are sequential–an individual's event status at time $t$ depends on their status at the previous time step $t-1$. In other words: once $\mathcal{N}(t-1) = 1$ (event occurred by time $t-1$), then $\mathcal{N}(t) = 1$; similarly, once $\mathcal{N}_C(t-1) = 1$ (censored by time $t-1$), then $\mathcal{N}_C(t) = 1$ and $\mathcal{N}(t) = 0$ (no event can occur after censoring). Furthermore, $\mathcal{N}(t) \to \mathcal{N}_C(t)$ captures the logical constraint that an event precludes censoring–if $\mathcal{N}(t) = 1$, then $\mathcal{N}_C(t) = 0$.

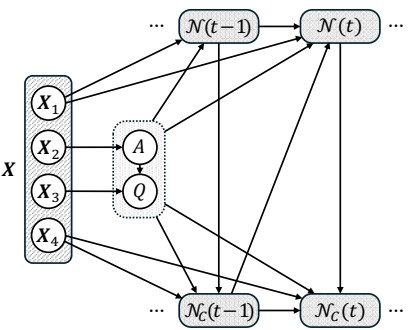

Figure G.1: Assumed DAG for our discrete-time survival analysis setting with continuous-valued treatment, extending prior formulations [14, 65].

