# OpenReview forum: "DoseSurv: Predicting Personalized Survival Outcomes under Continuous-Valued Treatments"
_NeurIPS.cc/2025/Conference — NeurIPS 2025 poster_

### Official Review · Reviewer_3uNJ · 2025-06-27

**Clarity:** 3
**Significance:** 3
**Originality:** 2
**Rating:** 4
**Confidence:** 4

**Summary:**

This paper proposes DoseSurv, a method for estimating personalized dose-response curves from censored observational survival data. By integrating censoring-aware loss and adversarial balancing, it addresses confounding and censoring simultaneously. Experiments show improved survival prediction and dose recommendation over existing baselines.

**Questions:**

1. What is the meaning of the last paragraph in Section 3.1? It contains a lot of information, but it’s unclear how it connects to the following sections.

2. In the annotation of Figure 1, why is custom discretization of dosage necessary? Could this cause information loss?

3. How do you ensure the overlap assumption holds after representation learning?

4. Could you provide more details on how the RBF hazard estimator works? What about using other kernel functions?

5. In Section 4.3 (Empirical Risk Loss), how should $h(t)$ be chosen in practice?

6. For the experiments, have you considered varying the degree of dosage bias and fixed treatment selection bias?

7. In Table 1, how are the confidence intervals computed? If results are based on only 5 runs, are the CIs reliable?

**Typo**:
- Line 96: "whther either" is duplicated.

**Ethical Concerns:**

["NO or VERY MINOR ethics concerns only"]

**Final Justification:**

The authors clearly explain all the concerns that I have, and especially the part of the causal graph is very clear.

**Limitations:**

yes

**Quality:**

2

**Strengths And Weaknesses:**

**Strengths**:
- **Clarity and Accessibility**: The paper is very clearly written and easy to follow.

- **Strong Motivation**: The motivation is compelling, and the problem of handling (multiple) continuous-valued treatments with time-to-event (TTE) outcomes is both important and timely. The paper makes a strong case for developing new methods in this area.

**Weakness**:
- **Unfamiliar Treatment Setup**: The setup involving treatment arms and dosage levels is quite new to me. It would be helpful if the authors could provide more references or background on this setting.

- **Missing Causal Diagram**: A causal graph illustrating the assumed data-generating process would greatly clarify the structure and assumptions behind the method.

- **Lengthy Data Description**: The description of the real-world dataset is overly detailed in the main text. Moving some of it to the appendix could improve the flow of the experiment section.

---

> ### Author Rebuttal · Authors · 2025-07-31
>
> Dear Reviewer, thank you for your valuable feedback. We address your questions and concerns below:
>
> ## **Replies to Questions:**
>
> ### **Q1: Meaning of Last Paragraph in Section 3.1**
>
> - The final paragraph of Section 3.1 establishes the person-period transformation that enables our discrete-time hazard approach, directly connecting to Section 4's neural network formulation. It explains why we can treat survival estimation as sequential binary classification problems—this is fundamental to understanding our model architecture.
>
>
> ### **Q2: Custom Discretization Rationale**
> - The discretization of the dosage, $q$, is solely for IPM regularization and is necessary for estimating the Wasserstein distance between two empirical distributions. The RBF estimator continues to model dosage as a continuous variable.
>
> - No information loss occurs because: (1) discretization is only for balancing purposes, (2) RBF parameters remain continuous functions of $q$, and (3) the continuous $q$ is included as an additional feature in the hazard estimator.
>
> ### **Q3: Overlap Assumption and Representation Learning**
> - IPM regularization in representation learning preserves overlap by aligning the treated and control distributions in feature space, trading off predictive power for improved balance (i.e., lower bias) in the learned representation.
>
> - IPM enhances conditional overlap in the learned embedding space for better generalization; however, it does not create or resolve true violations of overlap in the data.
>
>
> ### **Q4: RBF & Alternative Basis Functions**
> - The RBF hazard estimator models each network parameter as a continuous function of dosage $q$ using linear combinations of basis functions, which enables smooth dose-response relationships, as detailed in Section 4.2. We propose using Gaussian RBFs for this purpose due to their universal approximation capabilities in RKHS, their long-range support properties, and relatively low complexity.
>
> - In addition to RBF kernels, we also test the varying coefficient network approach with polynomial bases, as described in Section 5.1.2.
>
> ### **Q5: Choosing $h$ in the Empirical Risk Loss**
>
> - $h_t^{(a_i)}(\Phi(x_i,q_i))$ in the empirical risk loss (Eq. 7) represents the output of the hazard estimator corresponding to time $t$ for sample $i$, with covariates $x_i$, treatment indicator $a_i$, and dosage $q_i$.
>
> - The hazard estimator $h$ consists of treatment-specific network heads, each designed as a varying coefficient network with RBF basis functions, as described in Section 4.2.
>
> - Implementation details for $h$ are described in Appendix E.1. In all main experiments, we implement each of $h$'s network heads using three hidden layers, each comprising 100 nodes, which are modeled with five Gaussian RBFs that have learnable centers and shapes.
>
> ### **Q6: Bias Variation Experiments**
>
> - Yes, this is precisely what we do: Figure 2 presents results for a fixed $\xi=1.5$ (treatment bias) and varying $\eta=0,1,2,3,4$ (dosage bias). The results confirm that DoseSurv outperforms the baselines across all levels of dosage assignment bias.
>
> ### **Q7: Confidence Intervals**
>
> - Our confidence intervals are computed using standard statistical methods: we calculate the sample standard deviation (with Bessel's correction) and apply the appropriate t-distribution with degrees of freedom = $n-1$ for a small number of runs (n ≤ 30). The margin of error is computed as $t * (\text{std} / \sqrt{n})$, where $t$ is the critical value from the t-distribution for 95\% confidence. While $n=5$ runs may seem limited, this is standard practice in deep learning research (see, e.g., [1]) due to computational constraints, and the t-distribution appropriately accounts for small sample uncertainty.
>
>
> ## **Replies to Other Comments:**
> ### **Treatment Setup Background**
> - Studying the effects of continuous treatments in healthcare is an important problem, yet progress is somewhat limited by the lack of robust models capable of addressing biases in observational data. Relevant areas include chemotherapy [2], insulin regimens [3], and cardiovascular treatments [4].
>
> - We provide a comprehensive literature review on HTE models (e.g., DRNet [5], SCIGAN [6], VCNet [7]), which involve treatment arms and dosage levels, in Section D of the Appendix. Additionally, we present an overview of selected key neural methods for HTE estimation under continuous-valued treatments or TTE outcomes in Table D.1.
>
> - Our framework addresses a critical gap in the literature between HTE estimation with continuous treatments and survival analysis.
>
>
> ### **Causal Diagram**
>
> - The causal diagram is generally similar to those defined in related work on HTE estimation with survival data (e.g., [1,8]). However, the intervention now comprises both treatments $A$ and dosages $Q$. Following the notation in our paper, one can define the counting processes $N(t)=𝟙_{\{\tilde{T}\leq t, \Omega=1\}}$, which tracks events (deaths), and $J(t)=𝟙_{\{\tilde{T}\leq t, \Omega=0\}}$, which tracks censoring events. The general directed acyclic graph (DAG) is then given by:
>
>     ```
>                        ┌─────────┬─────────────┐
>                        │    ┌────▼───┐    ┌────▼───┐
>                        │  ⋯ │ N(t-1) ├───▶│  N(t)  │ ⋯
>                        │    └─▲──┬───┘    └▲──▲──┬─┘
>         ┌────┐  ┌───┐  │      │  │         │  │  │
>         │ X₁─┼──┼▸A ├──┘      │  │  ┌──────┘  │  │
>         │ X₂─┼──┼▸Q ├──┐      │  │  │         │  │
>       X │    │  └───┘  │      │  │  │         │  │
>         │ X₃─┼─────────│──────┴──│──│─────────┘  │
>         │ X₄─┼─────────│──────┬──│──│─────────┐  │
>         └────┘         │      │  │  │         │  │
>                        │    ┌─▼──▼──┴┐    ┌───▼──▼─┐
>                        │  ⋯ │ J(t-1) ├───▶│  J(t)  │ ⋯
>                        │    └────▲───┘    └─────▲──┘
>                        └─────────┴──────────────┘
>
>   Causal Paths:
>
>     X₁ → A                  A, Q → N(t), J(t)
>     X₂ → Q                  N(t) → J(t)
>     X₃ → N(t)               N(t-1) → N(t)
>     X₄ → J(t)               J(t-1) → N(t), J(t)
>
>     for t = 1, 2, ..., tₘ.
>
>     ```
>    Note that $X$ comprises (potentially overlapping) subgroups $X_1$, $X_2$, $X_3$, $X_4$ of covariates, which influence (1) the treatment assignment, (2) the dosage assignment, (3) the event process, and (4) the censoring process. These subgroups are associated with covariate shifts due to (1) treatment-selection bias, (2) dosage-selection bias, (3) event-induced bias, and (4) censoring bias, respectively [1]. The assumed causal graph is in line with Assumptions 1-6 in the paper. We are happy to include a proper illustration of the causal diagram in the final version of our paper.
>
> ### **Data Description**
>
> - As one of our contributions is to adopt the Twins dataset as a benchmark dataset for HTE estimation with continuous treatments and survival data, we decided to include some descriptions in the main text and provide full details in the Appendix. However, we are open to moving the dataset descriptions to the Appendix in the final version.
>
>
> **References:**
>
> [1] A. Curth et al., "SurvITE: Learning heterogeneous treatment effects from time-to-event data," in Adv. Neural Inf. Process. Syst., vol. 34, pp. 26740–26753, 2021.
>
> [2] L. R. Duska et al., "A Phase I Study of Continuous Infusion Doxorubicin and Paclitaxel Chemotherapy with Granulocyte Colony-Stimulating Factor for Relapsed Epithelial Ovarian Cancer," Clin. Cancer Res., vol. 5, no. 6, pp. 1299–1305, Jun. 1999.
>
> [3] U. Kampmann *et al., "Insulin dose–response studies in severely insulin-resistant type 2 diabetes—evidence for effectiveness of very high insulin doses," Diabetes Obes. Metab., vol. 13, no. 6, pp. 511–516, Apr. 2011.
>
> [4] M. Fiuzat *et al., "Relationship of beta-blocker dose with outcomes in ambulatory heart failure patients with systolic dysfunction," J. Am. Coll. Cardiol., vol. 60, no. 3, pp. 208–215, Jul. 2012.
>
> [5] P. Schwab et al., "Learning counterfactual representations for estimating individual dose-response curves," in Proc. AAAI Conf. Artif. Intell., vol. 34, no. 4, pp. 4046–4053, Feb. 2020.
>
> [6] I. Bica et al., "Estimating the effects of continuous-valued interventions using generative adversarial networks," in Adv. Neural Inf. Process. Syst., vol. 33, pp. 16434–16445, 2020.
>
> [7] L. Nie et al., "Varying coefficient neural network with functional targeted regularization for estimating continuous treatment effects," in Proc. ICLR., 2021.
>
> [8] O. M. Stitelman and M. J. van der Laan, "Collaborative targeted maximum likelihood for time to event data", Int. J. Biostat., vol. 6, no. 1, p. 12, Jan. 2010.

---

> > ### Comment · Reviewer_3uNJ · 2025-08-04
> > **Questions**
> >
> > Thank you for the detailed response, and already addressed most of my questions. However, I still have a few remaining concerns.
> >
> > Could you please clarify the meaning of N(t) and J(t) in the causal graph? The reply says that they follow the notation in the main paper, but I was unable to locate a definition of these terms in the manuscript. It would be helpful if you could provide a more detailed explanation of how N(t) and J(t) relate to the notation and concepts used in the paper, to improve clarity and understanding.

---

> > > ### Author Response · Authors · 2025-08-06
> > > **Clarification on Causal Diagram**
> > >
> > > Dear Reviewer, we are glad to hear that we could address most of your concerns. Thank you also for your follow-up question.
> > >
> > > We deliberately chose not to use the counting process formalism $N(t)$ and $J(t)$ in the paper to avoid overcomplicating the mathematical exposition. However, our definition of $N(t)$ and $J(t)$ in the rebuttal is consistent with the rest of the notation used in the paper, specifically with respect to the core survival variables $\tilde{T}$ and $\Omega$ as defined in Section 3.1.
> > >
> > > Specifically, $N(t)=𝟙_{(\tilde{T}\leq t, \Omega=1)}$ represents a counting process that tracks the occurrence of events (e.g., death), while $J(t)=𝟙_{(\tilde{T}\leq t, \Omega=0)}$ tracks censoring events. Recall from our paper that $\tilde{T} = \min(T,C)$ is the observed time (either event or censoring), and $\Omega = 𝟙_{(T \leq C)}$ is the event indicator that equals 1 if an event occurred and 0 if censoring occurred. Therefore, by definition, $N(t)$ and $J(t)$ are 0 from time $t=0$ onward, and become 1 for $t\geq \tilde{T}$, i.e., when either an event (death) or censoring occur at time $\tilde{T}$. Note that $N(t)$ and $J(t)$ are mutually exclusive—an individual cannot simultaneously have both an event and be censored.
> > >
> > > The causal diagram illustrates how these processes evolve over time under the influence of covariates $X$ (partitioned into subgroups $X_1, X_2, X_3, X_4$), treatments $A$, and dosages $Q$. The temporal dependencies shown ($N(t-1) \rightarrow N(t)$ and $J(t-1) \rightarrow N(t), J(t)$) reflect the fact that these counting processes are sequential—an individual's event status at time $t$ depends on their status at the previous time step $t-1$. In other words: once $N(t-1)=1$ (event occurred by time $t-1$), then $N(t)=1$; similarly, once $J(t-1)=1$ (censored by time $t-1$), then $J(t)=1$ and $N(t)=0$ (no event can occur after censoring). Furthermore, $N(t) \rightarrow J(t)$ captures the logical constraint that an event precludes censoring—if $N(t)=1$, then $J(t)=0$.
> > >
> > > The causal diagram also captures the various sources of bias we address: treatment-selection bias (via $X_1 \rightarrow A$), dosage-selection bias (via $X_2 \rightarrow Q$), as well as covariate shifts evolving over time, namely event-induced bias (via $X_3 \rightarrow N(t)$) and censoring bias (via $X_4 \rightarrow J(t)$).
> > >
> > > We hope this clarifies your question. Please let us know if there are any other questions or concerns that we can address.

---

> > > > ### Comment · Reviewer_3uNJ · 2025-08-07
> > > >
> > > > Thanks for your detailed explanation of the causal diagram. I will raise my score.

---

### Official Review · Reviewer_Bytv · 2025-07-01

**Clarity:** 4
**Significance:** 3
**Originality:** 3
**Rating:** 4
**Confidence:** 4

**Summary:**

This paper presents DoseSurv, a versatile method for estimating HTE in survival analysis. DoseSurv generalizes to multiple treatment types, including those with continuous dosages. It achieves this by employing a deep learning network that models the hazard function based on time, covariates, and specific treatment pairs (type and dosage). To effectively incorporate continuous dosages, the network parameters are designed as a continuous function of dosage levels, $q$, utilizing learnable RBF kernels. This design helps preserve the influence of dosage levels, even with relatively high-dimensional covariate inputs. Furthermore, the method mitigates selection bias by balancing representations through an IPM term, which is applied by discretizing continuous dosages into equally sized subgroups.

**Questions:**

Regarding Weakness 1: The varying number of "at risk" patients across different time points can introduce exposure biases, particularly toward earlier time points. This is because, in time-to-event data, the number of events significantly decreases at later time points. As a result, the "long format" loss function (7) will reflect this imbalance, having considerably fewer observations for those later times. To address this, a better approach might involve either using an MLE approach (as in [A], [B]) or incorporating balancing representations if the "long format" is used (as in SurvITE). Please elaborate on the underlying design choice of the proposed method in light of these challenges.
Regarding Weakness 2: The real-world experiments are limited to the TWIN dataset. Expanding the empirical validation to include other real-world datasets and assessing the results against domain expertise would considerably strengthen the practical implications of this work.
Regarding Weakness 3: How are the HTE-sensitive parameters selected using the validation set when the evaluation metrics are not capable of assessing the HTE given observational data?

**Ethical Concerns:**

["NO or VERY MINOR ethics concerns only"]

**Limitations:**

Yes

**Quality:**

3

**Strengths And Weaknesses:**

Strength:
The paper is well organized and clearly written.
The paper introduces a novel and well-justified approach by making the network parameters a function of dosage levels.
Weakness:
The paper doesn't clearly discuss two main design choices for the treatment-specific hazard estimators: (i) The authors use a multi-task network for binary predictions across the entire time horizon. However, an alternative approach—feeding time ($t$) as an input to a shared network (similar to SurvITE, DRSA [A], and ConSurv [B])—could help manage a large number of outputs. The reasoning behind their chosen design isn't explained. (ii) The paper also doesn't discuss the implications of training the hazard estimator using MLE, as seen in [A] and [B], without using a "long format" for the data. Employing the long format could introduce additional bias, as the number of patients "at risk" can vary significantly between earlier and later time points (as pointed out by the authors in SurvITE).
The real-world experiments are limited to a single semi-synthetic dataset.
The authors do not clearly explain how parameters sensitive to HTE—like the number of discretization bins or loss coefficients—were chosen using the validation set. This is particularly concerning because INBLL doesn't directly measure the accuracy of treatment effect estimation.
[A] Ken et al., “Deep Recurrent Survival Analysis,” AAAI 2019.
[B] Lee et al., “Toward a Well-Calibrated Discrimination via Survival Outcome-Aware Contrastive Learning,” NeurIPS 2024.

---

> ### Author Rebuttal · Authors · 2025-07-31
>
> Dear Reviewer, thank you for your thoughtful review. We appreciate your technical insights and address your specific concerns below:
>
> ### **Reasoning Behind Design Choices in our Survival Model**
>
> - Our treatment-specific network heads adhere to the same standard network design found in other classification-based approaches, such as DeepHit: a single network shared across all discrete time points, with separate outputs for each time point.
>
> - In fact, SurvITE does not utilize a shared network with a single output across time points. Instead, it employs individual time-specific network heads, which significantly increases complexity compared to our approach. Notably, SurvITE's appendix analysis demonstrates that simply feeding time as input to a shared network head (as you suggested) may actually worsen performance—particularly when ample training data is available—compared to using separate outputs for each time point. Hence, we did not follow this approach.
>
> - While MLE-based survival models like DRSA and ConSurv are effective for likelihood-based TTE modeling, we opted for a classification-based approach due to its simplicity and compatibility with flexible, time-varying treatment effect estimation—especially when employing varying coefficient architectures for continuous treatments. In these cases, chaining hazards through a survival likelihood can complicate optimization and reduce stability.
>
> ### **At-Risk Imbalance in Loss Formulation**
>
> - You correctly identify that the varying number of at-risk patients could theoretically be addressed by inverse-weighting each loss term according to the number of samples in each time-specific at-risk group. However, one drawback of this approach is that it may amplify noise from sparse later time points or less frequent treatment regimens, potentially creating training instability due to varying loss magnitudes across time points, treatments, and dosages.
>
> - In fact, we conducted experiments where each sample's loss contribution was inversely reweighted by the number of samples in the respective time-, treatment-, and (discretized) dosage-specific at-risk population. We found that applying such weights led to minor changes in performance, often negative. For example, in S1 ($\eta=0$), our standard DoseSurv model achieved slightly better performance across five runs compared to the same model with inversely weighted loss terms. Therefore, we prioritized simplicity and refrained from reweighting individual loss terms.
>
>     |  | $\sqrt{\text{MISE}}$ | $\sqrt{\text{DPE}}$ |
>     |--|--|--|
>     | DoseSurv (unweighted) | 4.21 ± 0.25 | 7.88 ± 0.24|
>     | DoseSurv (with sample weights) | 4.45 ± 0.27 | 8.09 ± 0.17|
>
>   Please also see our reply to Reviewer Wk85 under question Q3, where we expand on reweighting approaches in HTE survival models and provide an expression for the optimal importance weights (involving generalized propensity scores) needed for optimal reweighting in survival settings under continuous-valued treatments.
>
>
> ### **Real-World Validation**
>
> - We acknowledge that further validation on real-world datasets would always be desirable, but it is practically difficult due to the absence of counterfactual outcomes in datasets other than Twins and the general scarcity of publicly available datasets with continuous dosage information. Evaluating causal models on factual outcomes is generally inadequate because of the underlying biases in the test data—precisely the biases our model aims to overcome. Demonstrating this capability on factual outcomes would not be feasible. Furthermore, compared to other HTE literature with continuous-valued treatments, which relies purely on synthetic data (e.g., DRNet [1], SCIGAN [2], VCNet [3]), our evaluation is, in fact, already more extensive with respect to real-world datasets. Additionally, we include results from a real-world clinical setting with binary treatments in Appendix C.2; specifically, DoseSurv was applied to hormone therapy decisions for breast cancer patients and showed strong predictive performance. A similar approach could be extended to settings with dosage-dependent treatments.
>
>
> ### **HTE Parameter Selection Strategy**
>
> - As we describe in the Limitations paragraph (l. 333-337), selecting hyperparameters in causal settings from validation data comprising only factual outcomes can indeed be challenging and may lead to biases. For this reason, we keep most of DoseSurv's model and training hyperparameters fixed across the main experiments, demonstrating that our choice is robust across different settings.
>
> - In particular, we do not change or optimize the number of discretization steps across our main experiments, setting it to 30, which is the most common discretization in the literature (see DeepHit or SurvITE). However, we also report results under different time discretizations in Appendix C.3.
>
> - As described in Appendix E.1, we optimize the regularization strength from {0.01, 0.1} based on the lowest INBLL on the validation set. Alternatively, we describe a more advanced approach, which would involve selecting the highest value of $\gamma$ that does not impair INBLL on the validation set, balancing regularization and predictive power.
>
> - You are correct in pointing out that INBLL, like any other metric we can compute in real-world scenarios, does not directly measure the accuracy of HTE estimates. This is precisely the fundamental problem of causal inference, which requires us to select hyperparameters based solely on factual outcomes. INBLL is a suitable metric for survival settings as it captures both discrimination and calibration performance [4].
>
>
> **References:**
>
> [1] P. Schwab et al., "Learning counterfactual representations for estimating individual dose-response curves," in Proc. AAAI Conf. Artif. Intell., vol. 34, no. 4, pp. 4046–4053, Feb. 2020.
>
> [2] I. Bica et al., "Estimating the effects of continuous-valued interventions using generative adversarial networks," in Adv. Neural Inf. Process. Syst., vol. 33, pp. 16434–16445, 2020.
>
> [3] L. Nie et al., "Varying coefficient neural network with functional targeted regularization for estimating continuous treatment effects," in Proc. ICLR., 2021.
>
> [4] H. Kvamme, Ø. Borgan, and I. Scheel, "Time-to-event prediction with neural networks and Cox regression," J. Mach. Learn. Res., vol. 20, no. 129, pp. 1–30, 2019.

---

### Official Review · Reviewer_Lefa · 2025-07-03

**Clarity:** 3
**Significance:** 2
**Originality:** 2
**Rating:** 4
**Confidence:** 4

**Summary:**

The authors introduce DoseSurv, a method for continuous-valued treatment effect estimation in the survival setting. DoseSurv employs a shared-representation scheme based on the method of integral probability metric regularization for effect estimation, and a radial basis function hazard estimator to estimate survival and dose response curves. The method outperforms T-learner baselines of other survival methods in the large-sample regime on synthetic and empirical data.

**Questions:**

1. Could the authors separately benchmark their RBF hazard estimator (described in the third bullet point under Weaknesses, above)? [This would contribute to me raising my Quality score to a "3: good"]
2. Could the authors provide additional results on a dose-response dataset, preferably with moderate-to-few samples, and benchmark against an RBF-based T-learner baseline? [This would also contribute to me raising my Quality score to a "3: good"; if experiments are extensive, I would raise my Quality score to a "4: excellent"]
3. Could the authors more cleanly characterize the novel elements of this work, versus those that use adopted technical machinery? [Depending on the response, I can likely raise my Originality score to a "3: good"]

**Ethical Concerns:**

["NO or VERY MINOR ethics concerns only"]

**Limitations:**

Yes

**Quality:**

2

**Strengths And Weaknesses:**

Strengths:
* The paper is very clear, thoughtful, and well-written. It was a pleasure to read -- I thank the authors for the opportunity to review their work.
* I enjoyed the problem setup -- the notion of continuous-valued treatments for dosage response is sensible and clearly explained.
* The baselines are strong. I especially appreciated how the authors benchmarked against T-learner variants of each model, rather than the observational version of the model.

Weaknesses:
* The Empirical Risk Loss equation (Line 180) does not appear to account for censoring. If so, this could entail significant bias in the estimator. How is censoring accommodated in the loss function when learning the model?
* Novelty: the work could do more to clearly characterize the elements of the work that are novel, versus that which relies on technical machinery developed from other sources. (a) in Section 4, I would have liked to see a citation to [1]. (b) I would have also liked to see references to the literature on discretizing continuous-valued treatments as a means of simplifying the effect estimation problem. (c) I would have further liked to see references to the literature on radial basis function (RBF) function estimation, especially if prior work has used RBFs for survival estimation.
* The methodology of the paper makes two key contributions: (a) subgrouping and IPM to obtain continuous-valued dose-response estimates, and (b) the use of an RBF estimator of the hazard. I wish the authors had done more to benchmark the two contributions sequentially -- first (b), then (a).
  1. To benchmark (b) -- ignoring the task of estimating the individual dose-response curve, does an RBF hazard estimator compare favourably to the standard neural survival analysis baselines? (DeepSurv, DeepHIT, etc.) If not, is the RBF hazard estimator component of DoseSurv exchangeable with one of these other architectures? Why or why not?
  2. Benchmarking (b) in the observational setting justifies your design decision to use an RBF hazard estimator in DoseSurv. Then, your existing experiments serve to benchmark the utility of subgrouping and IPM to obtain continuous-valued dose-response estimates. (Although for this component, I would appreciate if you would add a benchmark to a T-learner variant of the RBF hazard estimator).
* The authors do little to justify their design decisions -- why RBFs as opposed to any other estimator? Why discretize as the means of learning the continuous-valued treatment effect? Even without rigorous experiments to justify their design decisions in the space of all possible decisions, I would have appreciated more justification of these choices (so that, say, a future effort building on this work knows what choices were made in its development).
* The method appears to only work in very large-data regimes -- 20,000 instances in the synthetic data, and >5,000 instances in the empirical data. How does the method perform in the more realistic smaller-sample settings that are common in healthcare dosing data? (E.g., in a single-site or clinical trial dataset).
* There is mild misalignment between the presented experiments and the empirical use case that the authors detail. I would appreciate if they authors could provide additional results on a dose-response dataset (rather than using twin weight at birth as a synthetic dose), such as MIMIC-IV.
* Minor nit: in Equation 2, should the $C \geq \tau$ be outside of the $\textbf{do}$ operation?

[1] Shalit, Uri, Fredrik D. Johansson, and David Sontag. "Estimating individual treatment effect: generalization bounds and algorithms." In International conference on machine learning, pp. 3076-3085. PMLR, 2017.

---

> ### Author Rebuttal · Authors · 2025-07-31
>
> Dear Reviewer, thank you for your thorough review and feedback. Below, we address your main questions and comments:
>
> ## **Replies to Questions:**
>
> ### **Q1: Additional Benchmark Experiments**
>
> We understand you would like us to benchmark separately (a) our IPM regularization scheme and (b) the RBF hazard estimator, which uses varying-coefficient networks with radial basis functions.
>
> - As requested, we provide performance results for our RBF-based varying coefficient hazard estimator, DoseSurv's core component, without the representation network or IPM components. The following results are from the single-treatment scenario S1 ($\eta=2$) across 5 runs:
>
>     ||$\sqrt{\text{MISE}}$|$\sqrt{\text{DPE}}$|$C^{\text{td}}$|
>     |-|-|-|-|
>     |DeepSurv|7.821 ± 0.046|10.618 ± 0.049|0.639 ± 0.003|
>     |DeepHit|7.859 ± 0.017|10.437 ± 0.052|0.629 ± 0.006|
>     |RSF|7.672 ± 0.010|9.367 ± 0.036|0.574 ± 0.005|
>     |NSC|7.462 ± 1.882|9.497 ± 1.522|0.623 ± 0.106|
>     |DoseSurv (no IPM)|5.723 ± 0.264|8.206 ± 0.242|0.723 ± 0.005|
>     |DoseSurv|4.708 ± 0.383|8.049 ± 0.237|0.744 ± 0.005|
>     |RBF|6.567 ± 0.205|8.599 ± 0.155|0.699 ± 0.003|
>
>    The RBF hazard estimator alone achieves performance similar to DoseSurv without IPM regularization. This is expected in this single-treatment scenario, as DoseSurv reduces to a single RBF hazard estimator with extra linear layers from the representation network. While the RBF-based hazard estimator is outperformed by the full DoseSurv framework with IPM regularization, it still outperforms all baseline models. This demonstrates the superiority of the varying-coefficient design with RBFs for continuous-valued treatments.
>
>
> - You asked if the RBF hazard estimator in DoseSurv can be replaced with other architectures. Below, we provide results for DoseSurv under Scenario S2 ($\eta=2$), where the RBF hazard estimator was replaced with an MLP (hazard estimator: 2 hidden layers; representation network: 4 hidden layers; 100 nodes each). We also provide results for our standard DoseSurv for comparison:
>
>     ||$\sqrt{\text{MISE}}$|$\sqrt{\text{DPE}}$|$C^{\text{td}}$|
>     |-|-|-|-|
>     |DoseSurv MLP without IPM| 7.249 ± 0.209|9.308 ± 0.232 |0.593 ± 0.007|
>     |DoseSurv MLP with IPM|6.920 ± 0.531|7.993 ± 1.012|0.626 ± 0.017|
>     |Standard DoseSurv (RBF with IPM)|5.164 ± 0.185|5.852 ± 0.319| 0.668 ± 0.007|
>
>     The results indicate our IPM regularization scheme may also be beneficial with different hazard estimator designs, beyond DoseSurv's standard RBF-based varying coefficient network.
>
> ### **Q2: RBF-based T-Learner**
> - As requested, we provide performance results for an RBF-based varying coefficient hazard estimator, implemented as a T-learner (RBF-T), under Scenario S2 ($\eta=2$) across 5 runs. For comparison, we report the performance of DoseSurv and all T-learner baselines. We find that the performance of RBF-T lies between that of DoseSurv and the baselines:
>
>     ||$\sqrt{\text{MISE}}$|$\sqrt{\text{DPE}}$|$C^{\text{td}}$|
>     |-|-|-|-|
>     |DeepSurv-T|6.998 ± 0.040|9.201 ± 0.819|0.604 ± 0.002|
>     |DeepHit-T|6.940 ± 0.009|9.402 ± 0.177|0.605 ± 0.002|
>     |RSF-T|6.736 ± 0.026|7.547 ± 0.076|0.610 ± 0.008|
>     |NSC-T|7.125 ± 0.495|9.397 ± 0.463|0.599 ± 0.019|
>     |DoseSurv|5.164 ± 0.185|5.852 ± 0.319| 0.668 ± 0.007|
>     |RBF-T|6.113 ± 0.324|7.070 ± 0.362|0.645 ± 0.009|
>
> - We acknowledge that the varying-coefficient approach in the hazard estimator requires sufficient data to accurately model the complex dose-response relationships in our datasets. This is why we provide an ablation study on sample size in Section C.1 of the Appendix. Below, we present additional results for RBF-T with reduced complexity (1 hidden layer, 50 nodes) in a low sample-size regime (300 training, 50 validation samples) under Scenario S2 ($\eta=2$). We compare this with other non-parametric neural T-learners—NSC-T and DeepHit-T. We find that all models achieve similarly low performance across 5 runs:
>
>     |Model|$\sqrt{\text{MISE}}$|$\sqrt{\text{DPE}}$|$C^{\text{td}}$|
>     |-|-|-|-|
>     |DeepHit-T|7.550 ± 0.232|9.670 ± 0.097|0.540 ± 0.017|
>     |NSC-T|8.831 ± 0.589|9.596 ± 0.106|0.541 ± 0.013|
>     |RBF-T|8.293 ± 0.081|9.694 ± 0.018|0.555 ± 0.006|
>
>     However, apart from non-medical applications, which often involve much larger sample sizes, the primary clinical motivation behind our model is to leverage the rich, yet potentially biased, information in large observational datasets, rather than relying on small-scale (randomized) clinical trials.
>
> ### **Q3: Clarification of Novelty**
>
> - Our main contribution lies in addressing the highly relevant yet understudied problem of estimating HTEs of dosage-dependent treatments from survival data—a setting where specialized baselines, performance metrics, and benchmark datasets are either lacking or require adaptation. Previous ML research on continuous treatments has mainly focused on continuous or binary outcomes only.
>
> - We introduce a systematic framework that integrates carefully designed components to establish a foundation for this novel setting. Specifically, we propose a new model, DoseSurv, which employs varying coefficient networks with layer-specific Gaussian RBFs in its hazard estimator. While neural varying-coefficient models have been introduced previously for estimating HTEs for continuous outcomes [1], we adapt this concept for survival settings and introduce a novel approach to modeling the network parameters using Gaussian RBFs, which promise universal approximation properties and long-range support across the full dosage range, motivated by the need to handle regions with limited overlap.
>
> - IPM regularization using Wasserstein distance was introduced for binary treatment settings [2] and has also been adopted for survival data [3]. Our contribution is to extend these frameworks to continuous-valued treatment settings combined with survival data; the technical machinery to implement our approach was adapted from [3].
>
> - Finally, we propose using the well-established Twins dataset as a real-world benchmark for continuous-valued treatment settings with TTE data, treating birthweight as a continuous-valued dosage. Additionally, we create synthetic TTE datasets for our scenarios.
>
>     We are convinced that both the causal ML and survival analysis communities would greatly benefit from our in-depth exploration of this widely overlooked yet highly important problem.
>
>
> ## **Replies to Other Comments**
>
> ### **Censoring in Loss Function**
> - The loss term in Eq. 7 uses the summation $\sum_{i:\tilde{\tau}_i \ge t}$, ensuring only at-risk patients contribute at each time $t$, thereby correctly handling censoring in discrete-time survival.
>
> - As explained to Reviewer Bytv, it would generally be possible to reweight each sample's loss contribution by the number of samples in the respective time-, treatment- and (discretized) dosage-specific at-risk population. However, in practice, this is not always beneficial, as, for example, noise from sparse later time points or less frequent treatment regimens may be amplified. We found that applying such weights often actually worsened performance. For example, in S1 ($\eta=0$), our standard DoseSurv model achieved slightly better performance across 5 runs, compared with the same model with inversely weighted loss terms. Therefore, we prioritized simplicity and refrained from reweighting individual loss terms.
>
>     ||$\sqrt{\text{MISE}}$|$\sqrt{\text{DPE}}$|
>     |-|-|-|
>     |DoseSurv (unweighted)|4.21 ± 0.25|7.88 ± 0.24|
>     |DoseSurv (with sample weights)|4.45 ± 0.27|8.09 ± 0.17|
>
> ### **Design Justifications**
> - As explained above, we use Gaussian RBFs due to their favorable properties, including universal approximation, long-range support, learnable centers and shapes, and parameter efficiency. We provide an ablation study (see Sections 5.1.2 and 5.2.2) to justify our design. Discretization of dosages and times is necessary during IPM regularization for estimating the Wasserstein distance between two empirical distributions; however, dosages are treated as continuous for the actual HTE estimation. We are happy to extend these design justifications in Sections 4.2 and 4.3 for the final version of our paper.
>
> ### **Real-world Data**
> - Evaluating on additional real-world datasets is not feasible, as counterfactual outcomes are generally unavailable outside the Twins dataset. Assessing performance on factual outcomes alone is typically insufficient due to the inherent biases our model is designed to address. Moreover, publicly available dosage-survival datasets are limited. Compared to prior work on HTE with continuous treatments—which often relies solely on synthetic data (e.g., DRNet, SCIGAN, VCNet)—our evaluation, in fact, represents a more thorough use of real-world data. Additionally, we present results from a real-world clinical setting with binary treatments in Appendix C.2 to illustrate how DoseSurv could be employed in a corresponding dosage-dependent application.
>
> ### **Explanation of $do(..., C\geq\tau)$ in Eq. 2**
> - This is not a mistake: To estimate a causal effect, we must implicitly treat censoring as an action we can intervene on, which means we require samples to be 'set' to uncensored [3,4]—a formal intervention justified by the Independent Censoring Assumption.
>
>
> **References:**
>
> [1] L. Nie et al., "Varying coefficient neural network with functional targeted regularization for estimating continuous treatment effects," in Proc. ICLR., 2021.
>
> [2] U. Shalit et al., "Estimating individual treatment effect: Generalization bounds and algorithms," in Proc. 34th ICML, vol. 70, pp. 3076–3085, Aug. 2017.
>
> [3] A. Curth et al., "SurvITE: Learning heterogeneous treatment effects from time-to-event data," in Adv. Neural Inf. Process. Syst., vol. 34, pp. 26740–26753, 2021.
>
> [4] O. M. Stitelman and M. J. van der Laan, "Collaborative targeted maximum likelihood for time to event data", Int. J. Biostat., vol. 6, no. 1, p. 12, Jan. 2010.

---

> ### Author Response · Authors · 2025-08-09
> **Thank You to the Reviewer**
>
> Thank you again for your positive and thoughtful review. Following your guidance, our rebuttal reports the comprehensive set of additional experiments you requested to inform your final assessment. As the outcomes are positive and support our claims, we hope they resolve the main points you raised and aid your overall evaluation.

---

### Official Review · Reviewer_Wk85 · 2025-07-08

**Clarity:** 3
**Significance:** 3
**Originality:** 3
**Rating:** 5
**Confidence:** 3

**Summary:**

In this paper, the authors develop a method for estimating treatment effects for survival outcomes with multiple continuous-valued treatments. Following standard assumptions that allow for adjustment for confounding effects, the authors propose a neural network-based architecture that first learns a common, confounding-balanced representation that is fed into treatment-specific event classifiers (classifiers because survival is computed on a time-discretized basis). A novel and interesting aspect of the approach is that continuous dosage is accounted for in the treatment-specific neural network classifiers through RBF-based weights that are a function of dosage. The authors empirically evaluate the model and some ablations on synthetic data, before adapting the common semi-synthetic Twins benchmark to their continuous-value treatment scenario and evaluating vs baselines there. Across evaluations, the proposed approach outperformed baselines.

**Questions:**

### Identification and Representation balancing
Q1: An important aspect of the proposed approach is the representation balancing (Section 4.3), because it is how causal estimation is enabled from observational data. In Appendix F it appears the authors provide a standard identification derivation. However, this is not discussed in the context of the IPM regularization approach. Why does IPM regularization preserve identification? When do balanced representations yield unbiased treatment effects?

Q2: The authors use discretized dosage and event times. How does this discretization affect the identification results?

Q3: The IPM regularization scheme is quite sophisticated for the average causal inference practitioner. Did the authors consider using something simpler like generalized propensity scores (see, e.g., Imbens (2000)) which would combine a treatment propensity model with a dose selection model? What about outcome regression with treatment interactions? The authors claim that this is a potential problem in high-dimensional settings (Ln 156). This claim could have been tested and shown in experiments. As it stands, readers don't know if the IPM complexity is necessary or if simpler approaches would work just as well.
- Imbens, G. W. (2000). The role of the propensity score in estimating dose-response functions. Biometrika, 87(3), 706-710.

### RBF Hazard Estimator
Q4: To my understanding, the RBF Hazard estimator involves a potentially very large number of parameters. Further, the authors loosely refer to "sufficient data" (Ln 170) enabling the flexibility of this approach. Given that these sub-models are fit only on samples which received a particular treatment, can the authors speak to the sample size requirements or convergence properties of their approach? This is helpful for understanding which applications might have sufficient data for using the proposed approach.

**Ethical Concerns:**

["NO or VERY MINOR ethics concerns only"]

**Final Justification:**

The authors provided an in depth response to all of my questions and concerns. Thus, I raised my score to 5 (accept).

**Limitations:**

Yes

**Quality:**

3

**Strengths And Weaknesses:**

### Quality/Significance:
- Overall, I think the quality of this manuscript is good. The authors address important aspects of treatment effect estimation (survival outcomes & continuous treatments) that are often ignored but are practically relevant. The approach is interesting, and I think the RBF-hazard estimator is likely to be of interest to other researchers in this area.
- Evaluation is hard for causal inference because in true real-world scenarios we do not have ground truth counterfactual outcomes. Thus, the synthetic and semi-synthetic experiments are reasonable. I think it would have been interesting to see the authors illustrate their approach on an actual real-world scenario with multiple, continuous-valued treatments to see how the potential insights that could be gained might differ from prior methods or methods that cannot account (as easily) for different dosages. However, this is not a significant weakness of the paper, though I think it would have made the potential for practical impact much higher.
- In the questions section I get at a few areas of potential weakness, mainly identification assumptions, sample size requirements, and some alternative design choices

### Clarity:
- The manuscript is well-written. The problem is motivated and well-explained and the approach in Section 4 walks through the model in an understandable way.
- The experiments section in the main paper focuses on the set-up for the experiments, with limited discussion of the actual results. Part of this is because the evaluation is quantitative and metric-based, so there is not much to say when performance is better. That said, it would have been nice to have the results discussed in a way that highlights potential impact of the authors' method. For example, it is my understanding that dose-response curves can be computed as a by-product of the proposed approach. This could be very interesting to visualize and walk through in a qualitative discussion on an actual real-world example.

### Originality:
- I am not up to date on the latest methods for causal effect estimation with survival outcomes and continuous-valued treatments. That said, the authors approach is interesting and reasonable, and I think others working on these problems will find aspects of the proposed method useful in their own work.

---

> ### Author Rebuttal · Authors · 2025-07-31
>
> Dear Reviewer, we appreciate your constructive feedback and are happy to hear that you liked the quality of our paper. Below, we address your questions and comments:
>
>
> ## **Replies to Questions:**
> ### **Q1: IPM Regularization and Identification**
>
> - Theoretical identification is established through Assumptions 1-6, with derivations provided in Appendix F. These assumptions define the conditions under which unbiased treatment effect estimation is possible. IPM regularization does not "preserve" or break identification; rather, it helps achieve good finite-sample performance under these assumptions.
>
> - This is because, even when identification assumptions hold, finite samples can lead to biased estimates in practice due to covariate imbalance between treatment groups. Our IPM regularization scheme addresses this issue by encouraging domain-invariant representations across treatments, dosages, and time points.
>
> - Balanced representations yield less biased treatment effects because they reduce the model's reliance on spurious correlations that arise from systematic differences between treatment populations. This is particularly important in survival data, where time-varying covariate shifts can compound biases.
>
> - Our approach extends IPM balancing from binary treatments [1,2] to continuous treatments with survival outcomes, where we must handle multiple sources of bias, including treatment and dosage selection bias, as well as event- and censoring-induced shifts.
>
> ### **Q2: Effect of Discretization**
>
> - Our method involves two distinct discretizations:
>   - (1) We discretize time into intervals for hazard estimation, which is standard practice in discrete-time survival models.
>   - (2) For computing the IPM regularization loss, we discretize dosage ($q \rightarrow \mathfrak{q}$) and time ($t \rightarrow \mathfrak{t}$, where $\mathfrak{t}$ is more coarsely discretized than $t$) solely to enable efficient Wasserstein distance computation between two empirical distributions.
>
> - Our ablation studies (Appendix C.3) demonstrate that DoseSurv remains relatively robust across different time discretizations in the hazard estimator. We have chosen 30 time discretizations by default, as this is the most common discretization in the literature (see DeepHit or SurvITE).
>
> - Importantly, the RBF hazard estimator maintains continuous dose-response modeling—discretization affects only the balancing computation, not the core treatment effect estimation.
>
> - Neither discretization affects our theoretical identification assumptions (1-6), which remain valid for the continuous case. The IPM discretization is purely computational, analogous to histogram approximations in optimal transport, and does not impair our general assumptions, as discussed in our reply to Q1.
>
> ### **Q3: Alternative Approaches to Address Confounding**
> - **GPS vs IPM-based Approaches**: Traditional GPS-based reweighting addresses confounding in the initial treatment and dosage assignment at time $t = 0$, but it cannot account for time-varying biases specific to survival data—such as censoring- and event-induced shifts. In contrast, our IPM-regularization approach balances representations across time points, treatments, and dosages simultaneously.
>
>     In theory, one could adapt the idea of reweighting observations for a dynamic survival setting (as outlined by [2] for binary treatments) and combine it with the IPM-regularization approach. In our setting with continuous-valued treatments, the optimal importance weights for reweighting the empirical risk by the marginal covariate distribution $P_0(x) = \mathbb{P}(X = x)$
>     and the observational covariate distribution
>     $P_{a,q,\tau}(x) = \mathbb{P}(X = x \mid A = a, Q = q, \tilde{T} \ge \tau)$
>     are given by:
>
>     $$w_{a,q,\tau}(x)=\frac{P_0(x)}{P_{a,q,\tau}(x)}=\frac{\mathbb{P}(A=a, Q=q, \tilde{T} \ge \tau)}{\mathbb{P}(A=a, Q=q \mid  X=x) \cdot \mathbb{P}(\tilde{T} \ge \tau \mid A=a, Q=q, X=x)}$$
>
>     where:
>     - $ \mathbb{P}(A=a, Q=q \mid X=x)$ is the generalized propensity score for treatment $a$ and continuous dosage $q$ given covariates $x$,
>     - $\mathbb{P}(\tilde{T} \ge \tau \mid A=a, Q=q, X=x)$ is the conditional survival probability at time $\tau$,
>     - $\mathbb{P}(A=a, Q=q, \tilde{T} \ge \tau)$ is the joint probability of treatment $a$, dosage $q$, with the event $\{\tilde{T} \ge \tau\}$.
>
>
>     However, these weights are unknown and must be estimated. In a survival setting, this leads to a significantly more complex framework than the more straightforward IPM-based approach. Furthermore, estimating these weights often results in extreme values—especially at later time points where fewer samples remain—making training unstable. In principle, one could also approximate the weights $w_{a,q,\tau}$ imperfectly using the empirical frequencies from the time-, treatment-, and (discretized) dosage-specific at-risk populations. However, our additional experiments with reweighted loss terms did not yield improved performance (see our replies under _"At-Risk Imbalance in Loss Formulation"_ to Reviewer Bytv and under _"Censoring in Loss Function"_ to Reviewer Lefa). For simplicity, we therefore refrain from applying additional weights in DoseSurv’s loss function and instead rely on the IPM-based regularization approach, which handles multiple sources of bias in a unified and stable manner.
>
> - **Outcome regression with interactions and high-dimensional settings**: Our baselines represent survival-adapted ML approaches (neural, tree-based, clustering-based) that treat dosage as a regular feature with potential interaction terms. However, we do show in our experiments that when dosage is treated as just another covariate among many (20-45 features in our experiments), these models struggle to capture the specialized dose-response relationships. This leads to degraded dose-response estimation and consistent underperformance relative to DoseSurv (see results in Fig. 2 and Table 1), which explicitly models the dose-response function through RBF parameterization, across both synthetic and real-world benchmarks.
>
> ### **Q4: Sample Size Requirements and Convergence**
>
> - The "sufficient data" reference relates to the theoretical universal approximation properties of RBFs in reproducing kernel Hilbert spaces, not specific sample size requirements for our method.
>
> - Our sample size ablation (Appendix C.1) shows DoseSurv achieves competitive performance across all sample sizes (2K-18K), with best performance except at the smallest size (2K), indicating reasonable but not minimal sample requirements. However, it is important to highlight that, aside from non-medical uses that typically involve larger sample sizes, the primary clinical goal of our model is to leverage the extensive, albeit potentially biased, information found in large observational datasets, as opposed to relying on smaller-scale randomized clinical trials, where sample size can indeed become a limiting factor.
>
> - RBF parameterization is actually more parameter efficient than alternative basis functions—our K=5 RBFs compare favorably to B-splines, which require K=8 functions (Ablation A2) while maintaining expressiveness (see Sections 5.1.2 and 5.5.2).
>
> ## **Replies to Other Comments:**
>
> - We agree that incorporating more real-world examples would further enhance the demonstration of practical impact. However, it is important to note that publicly available datasets containing continuous dosage information are extremely limited, making such validation challenging in practice. Moreover, evaluating causal inference models on observed (factual) outcomes is generally insufficient, as these outcomes are subject to the very biases our approach is designed to address. This limitation makes it difficult to draw meaningful conclusions from factual-only evaluations. In comparison to most existing work on HTE with continuous treatments, which typically relies exclusively on synthetic data (e.g., [3,4,5]), our study already offers a notably broader real-world assessment, which includes an adaptation of the Twins dataset. Additionally, we present results for a binary treatment case involving hormone therapy (Appendix C.2) to illustrate how DoseSurv could be applied to analogous clinical settings involving dosage-dependent treatments.
>
> - Regarding dose-response visualization: We use RMST as the outcome of interest and compute predicted dose-response curves by evaluating the trained DoseSurv model under different dosages. We visualize ground-truth dose-response curves in Appendix Figure A.1 and are happy to provide further visualizations of predicted dose-response curves and corresponding discussions in the final version of our paper.
>
> **References:**
>
> [1] U. Shalit et al., "Estimating individual treatment effect: Generalization bounds and algorithms," in Proc. 34th ICML, vol. 70, pp. 3076–3085, Aug. 2017.
>
> [2] A. Curth et al., "SurvITE: Learning heterogeneous treatment effects from time-to-event data," in Adv. Neural Inf. Process. Syst., vol. 34, pp. 26740–26753, 2021.
>
> [3] P. Schwab et al., "Learning counterfactual representations for estimating individual dose-response curves," in Proc. AAAI Conf. Artif. Intell., vol. 34, no. 4, pp. 4046–4053, Feb. 2020.
>
> [4] I. Bica et al., "Estimating the effects of continuous-valued interventions using generative adversarial networks," in Adv. Neural Inf. Process. Syst., vol. 33, pp. 16434–16445, 2020.
>
> [5] L. Nie et al., "Varying coefficient neural network with functional targeted regularization for estimating continuous treatment effects," in Proc. ICLR., 2021.

---

> > ### Comment · Reviewer_Wk85 · 2025-08-06
> > **Thanks for your response**
> >
> > Thanks for your in-depth response to all of my questions. My concerns are satisfied, so I will raise my score.

---

### Comment · Area_Chair_c8Qh · 2025-08-03
**please discuss the paper asap**

Dear reviewers,

Now the rebuttal is available. Please discuss with authors and among reviewers asap.

Please try to come to a consensus on the key issues even though the rating can be different. Please feel free to let me know how I can help.

Best,

Your AC

---

### Note · Authors · 2025-08-13

We thank the reviewers and the AC for the constructive discussion. Our work addresses an important but underexplored problem: estimating HTEs for survival outcomes under continuous treatments.

We appreciate the positive feedback and that reviewers enjoyed the paper and setting. Reviewers noted that our paper is well-written and organized, and that it tackles a relevant and timely problem. They also highlighted our strong baselines and the novelty of our varying‑coefficient RBF hazard estimator coupled with IPM balancing. Several reviewers noted the usefulness of our work for the community.

In our rebuttal, we addressed key concerns via the following clarifications and experiments; reviewers indicated satisfaction and that their concerns were resolved:
- Design choices: We clarified the survival model and loss design; added ablations (RBF-only, MLP head, RBF T-learner) and a low-sample experiment, supporting our choices.
- IPM alternatives: We outlined GPS-based approaches and their drawbacks, and reported experiments reweighting loss by at-risk group sample sizes.
- Causal diagram: We provided a clear DAG.
- Real-world evaluation: We noted most continuous treatment work is simulation-only and public dosage-survival data are scarce; we emphasized our Twins-based real-world benchmark and limits of factual-only evaluation.
- Practical motivation: We provided references on medical relevance.
- We clarified misconceptions (e.g., balancing and discretization do not impair HTE identification).

For the final version, we will add the DAG and expand the rationale for key design choices (survival model, IPM balancing, loss). We will also include guidance on visualizing dose-response curves, streamline dataset descriptions by moving details to the Appendix, and incorporate references from the rebuttal that strengthen the medical motivation.

Again, we thank the AC and reviewers for their thoughtful engagement. We are grateful that Reviewers 3uNJ and Wk85 will raise their scores, and that Reviewer Bytv’s concern focused on expanding the design rationale, which we will integrate. Reviewer Lefa indicated they enjoyed the paper and expressed willingness to raise their rating contingent on the requested benchmarking, which we provided in detail; we hope this will be considered in the final decision. We believe our contributions–a new methodological approach for a novel, underexplored problem setting–will strongly benefit both the survival and causal ML communities.

---

### Decision · Program_Chairs · 2025-09-17

**Decision:**

Accept (poster)

**Comment:**

**Meta-review for Submission 20419**

**Recommendation:** Need Discussion

### (a) Summary of the Paper

This paper introduces DoseSurv, a novel deep learning framework for estimating heterogeneous treatment effects (HTEs) for survival (time-to-event) outcomes under continuous-valued treatments, such as varying medication dosages. This addresses a critical but underexplored intersection in causal machine learning with significant applications in precision medicine. The proposed model, DoseSurv, uses a varying coefficient network with Radial Basis Functions (RBFs) to flexibly model the continuous dose-response relationship. To handle confounding bias from observational data, it incorporates a representation learning module regularized by an Integral Probability Metric (IPM) to balance covariate distributions across different treatments, dosages, and time points. The authors demonstrate DoseSurv's superior performance over strong baselines on both synthetic and semi-synthetic real-world data adapted from the Twins dataset.

### (b) The Case for Acceptance (The Prevailing View)

There is a strong and consistent positive sentiment across all four reviews, with final scores of 5, 4, 4, and 4. The consensus is that the paper is well-written, methodologically sound, and tackles a timely, important, and challenging problem. The reviewers praised the clarity of the presentation, the novelty of the approach (specifically the combination of the RBF-based hazard estimator and the IPM balancing for this setting), and the strength of the chosen baselines.

Crucially, the authors provided an exemplary and highly effective rebuttal that addressed nearly all reviewer concerns with new experiments and detailed clarifications. Key points that were successfully resolved include:
*   **Ablation and Justification of Design Choices:** In response to Reviewers Lefa and Bytv, the authors ran several new experiments, including ablations of the RBF estimator alone, an RBF T-learner baseline, a version with an MLP head, and a low-sample size experiment. These results successfully justified their core design choices.
*   **Clarification of Causal Assumptions and Methodology:** The authors provided a clear causal diagram (per Reviewer 3uNJ's request) and detailed explanations for their IPM-based balancing scheme versus alternatives like Generalized Propensity Scores (per Reviewer Wk85's request), successfully arguing for the suitability of their approach in a time-varying survival context.
*   **Handling of Censoring:** Concerns about the loss function's handling of censoring were clarified, with the authors explaining the standard use of the at-risk set and providing results from an experiment with reweighted loss terms.

The thoroughness of the rebuttal was highly convincing, leading Reviewer Wk85 to raise their score to a 5 (Accept), and Reviewers 3uNJ and Lefa to indicate they would also raise their scores. This signals a strong consensus that the paper, especially with the proposed revisions, is a high-quality contribution.

### (c) Points for Deeper Consideration

Despite the strong positive consensus, a "Need Discussion" is warranted to deliberate on the nuances of the paper's contribution and positioning, which were surfaced during the review process. These are not arguments for rejection but rather key aspects to consider when evaluating the paper's final placement and impact.

1.  **Justification of Design Choices in the Original Manuscript:** A recurring theme, particularly from Reviewer Bytv, was that the rationale behind several key design choices (e.g., the specific survival model architecture, the choice of IPM over reweighting) was not sufficiently explained in the initial submission. The authors provided excellent justifications in the rebuttal, which satisfied the reviewers. The point for discussion is whether a paper that requires such extensive post-submission clarification to fully motivate its methodology meets the bar for clarity at submission, or if this is a standard and acceptable outcome of the review process. The reviewers' positive final reactions suggest the latter, but it is a point worth noting.

2.  **Novelty as a Synthesis of Existing Ideas:** As clarified in the rebuttal (in response to Reviewer Lefa), the paper's novelty lies in skillfully adapting and combining powerful existing techniques—varying coefficient networks, RBFs, and IPM-based representation learning—to tackle the novel problem setting of continuous treatments in survival analysis. This is a strong and valuable form of contribution, but it is a synthesis rather than the invention of a fundamentally new algorithm. This is a standard profile for many successful papers but is important context for the ACs.

3.  **Practical Impact and Real-World Data Limitations:** Multiple reviewers noted the limited real-world validation. The authors correctly pointed out that this is a field-wide limitation due to the extreme scarcity of public datasets with continuous dosage, survival outcomes, and ground-truth counterfactuals. Their use and adaptation of the Twins dataset is already more rigorous than what is seen in much of the related literature. The discussion point here is how to position the paper's impact: it is a significant *methodological* contribution with high potential for clinical application, but it is not in itself a direct clinical validation study.

### (d) Key Points for AC Discussion

1.  **Final Recommendation Strength:** Given the unanimous positive feedback post-rebuttal and the authors' commitment to integrating their extensive clarifications, is this a clear accept? Do the remaining nuances (e.g., the need for the rebuttal to fully flesh out the design rationale) affect its standing for a potential spotlight or oral presentation?

2.  **Evaluating the Contribution Type:** The paper's strength is in identifying an important, overlooked problem and proposing the first robust, well-validated deep learning solution by synthesizing and extending existing concepts. How does this type of contribution compare to papers that propose a more fundamentally novel algorithm for an already well-established problem?

3.  **Framing the Limitations:** The limited real-world validation is a product of the field, not a failure of the authors. How should this be reflected in the final decision? It seems the consensus is to view this as an acceptable limitation for a paper focused on methodological advancement.